# The ACCESS-CM2 climate model with a higher resolution ocean-sea ice component (1/4°)

Wilma G. C. Huneke [1,2], Andrew McC. Hogg [1,2,3,4], Martin Dix [4,5], Daohua Bi [5], Arnold Sullivan [5,6], Shayne McGregor [3,6], Chiara M. Holgate [1,3], Siobhan P. O'Farrell [5], and Micael J. T. Oliveira [4]

[1]Research School of Earth Sciences, Australian National University, Canberra, Australia
[2]Australian Research Council Centre of Excellence for Climate Extremes
[3]Australian Research Council Centre of Excellence for the Weather of the 21st Century
[4]ACCESS-NRI, Australian National University, Canberra, Australia
[5]CSIRO Environment, Aspendale, Australia
[6]School of Earth Atmosphere and Environment, Monash University Melbourne, Victoria, Australia

**Correspondence:** Wilma Huneke (wilma.huneke@anu.edu.au)

**Abstract.** A new configuration of the Australian Community Climate and Earth System Simulator coupled model, ACCESS-CM2, with a higher resolution ocean-sea ice component at 0.25° is introduced. The higher resolution ACCESS-CM2-025 model was developed to better represent the ocean mesoscale and expand the scope of climate modelling research applications. The individual model components have not been changed compared with ACCESS-CM2-1, the existing lower resolution

version of the model at 1°, which was one of Australia's contributions to the World Climate Research Program's Coupled Model Intercomparison Project Phase 6 (CMIP6). This paper assesses the simulated climate for a 500 year present-day run in ACCESS-CM2-025 against observations, the lower resolution ACCESS-CM2-1 model, and two ocean-sea ice models using the same model components and comparable grid resolutions but with prescribed atmospheric forcing. ACCESS-CM2-025 is more energetic and performs better in regions of elevated ocean mesoscale variability such as at western boundary currents.

The higher resolution ACCESS-CM2-025 also features a more realistic ENSO life cycle and seasonality, with a reduced biennality, which is common in the lower resolution ACCESS-CM2-1. Both ACCESS-CM2 models share many biases, particularly near the sea surface and also affecting sea ice coverage, reflecting insufficiency in the atmospheric model component. While ACCESS-CM2-025 exhibits improved time-mean deep convection, sea ice, and mixed layer depth in the North Atlantic, it also experiences multidecadal variability, which is evident in many variables, including the Atlantic Meridional Overturning

Circulation.

## 1   Introduction

Coupled models are the primary tool to study the global climate system and to predict the ways in which it may change. They simulate the inherently coupled Earth System and thereby help to understand the governing physical processes and natural climate variability. They can also be applied to model past and future climates. For example, modelling centres around

the globe took part in the Coupled Model Intercomparison Project Phase 6 (CMIP6) to predict how the climate system will respond to changes in natural and anthropogenic forcing in the decades to come (Eyring et al., 2016). The accuracy of such

projections strongly depends on the quality of the employed coupled models which are known to have major biases (e.g., Notz, 2020; Beadling et al., 2020; Heuzé, 2021).

Coupled models combine a number of individual model components (typically including the atmosphere, land surface, ocean, and sea ice) and directly simulate the interaction between them. This interaction between the different model components allows the simulated climate system to freely evolve in time with only few assumptions on prescribed boundary conditions. However, coupled models are therefore very complex and heavily constrained by computational limitations. Computational considerations influence each model configuration, mainly the choice of parameterisations of physical processes as well as lateral grid resolution, with consequences for the simulated climate. Fortunately, computational capabilities steadily improve and provide the opportunity to advance specific aspects. Increasing the lateral grid resolution of the ocean model has a high priority as the ocean is a key component that links different parts of the climate system. A higher ocean grid resolution is not only expected to allow for a more realistic representation of the ocean (mean state and variability) but also impact how well sea ice and the lower atmosphere are simulated.

The ocean grid resolution affects not only geometric constraints on the representation of the coastline, passages, narrow straits, and narrow coastal shelves, but also impacts the ability of a model to simulate mesoscale dynamics. The explicit representation of the ocean mesoscale, such as eddies but also boundary currents and fronts, improves both the ocean mean state and climate variability as detailed in the following paragraphs. The Rossby deformation radius provides a length scale for the size of mesoscale features (Chelton et al., 1998). Eddies vary in size from a few kilometers to about 100 km depending on the latitude (smaller at high latitudes), ocean depth (smaller at shallow seas) and stratification (smaller at low stratification). A lateral grid resolution of less than half the Rossby deformation radius is needed in order for a model to simulate mesoscale features (Hallberg, 2013; Stewart et al., 2017). Based on the ability to represent the ocean mesoscale, models are classified as being either eddy-parameterising, eddy-present (eddy-permitting), or eddy-rich (eddy-resolving).

The majority of the CMIP6 models are eddy-parameterising while most of the existing eddy-present and eddy-rich models are part of the High Resolution Model Intercomparison Project (HighResMIP) (Haarsma et al., 2016). Eddy-parameterising models have a grid resolution of approximately 50-100 km (nominal resolution of $0.5°$ to $1°$) and require, as the name suggests, an eddy parameterisation such as the commonly employed Gent and McWilliams (GM) parameterisation (Gent and McWilliams, 1990). Eddy-present models have a grid resolution in the order of 25 km (nominal resolution of $0.25°$) and therefore resolve the Rossby deformation radius in lower latitudes but not at high latitudes or shallow (shelf) seas. These models lie in an in-between state and modelling centres have different approaches to the issues of whether and how to parameterise the unresolved ocean mesoscale while not overdamping the resolved eddies. Eddy-rich models are in the order of 10 km (nominal resolution of $0.1°$) or higher and are able to explicitly simulate the ocean mesoscale across most of the globe. However, even eddy-rich models do not resolve eddies poleward of approximately $60°$ (Hallberg, 2013). Global eddy-rich configurations are common for ocean-only models, but only very few coupled atmosphere-ocean models exist at this resolution (e.g. Roberts et al., 2019; Small et al., 2014; Sein et al., 2018).

Models that parameterise eddies are generally too diffusive and exhibit poor performance in regions of high eddy activity which affects various aspects of the climate system. For example, western boundary currents are too weak and wide in eddy-

parameterising models and their position is improved when the mesoscale is explicitly resolved (Small et al., 2014; Griffies et al., 2015; Sein et al., 2018). The Atlantic Meridional Overturning Circulation (AMOC), a measure of meridional volume transport in the Atlantic, is notoriously challenging to simulate as it is impacted by multiple processes on small scales including boundary currents, deep convection, and overflows (Jackson et al., 2023). The AMOC is stronger for higher resolution models, matches better the observed AMOC strength at 26° N, and has a larger latitudinal variation (McCarthy et al., 2015; Hirschi et al., 2020; Roberts et al., 2020). Similarly, the associated heat transport from low to high latitudes is stronger for high resolution models with a noticeable jump from eddy-parameterising to eddy-present models. The position and strength of the currents in the Nordic and Labrador Seas improve from eddy-parameterising to eddy-present to eddy-rich models including more realistic representations of the deep currents and overflows (Talandier et al., 2014; Colombo et al., 2020). In the Southern Hemisphere, the transport of the Antarctic Circumpolar Current (ACC), the strongest ocean current on Earth, is with its many individual fronts also sensitive to the ocean grid resolution (Hewitt et al., 2020). The explicit representation of eddies in the Southern Ocean is of particular relevance for the ACC strength in a changing climate as they balance increased momentum input by the westerlies through a phenomenon known as eddy saturation that is not captured by eddy parameterisations (Meredith and Hogg, 2006; Munday et al., 2013). Models in the eddy-present regime poorly represent the mesoscale in this latitude range where their ability to explicitly resolve the mesoscale is limited (Hallberg, 2013).

A finer ocean resolution not only improves the ocean physics but also impacts the atmosphere in coupled models through heat, freshwater, and momentum fluxes. Traditionally, air-sea fluxes have been considered from the atmosphere to the ocean on larger scales where the ocean suppresses atmospheric high-frequency variability and favours a low-frequency response (Hasselmann, 1976; Frankignoul and Hasselmann, 1977). However, observations show enhanced small-scale features in the wind pattern over regions of the ocean with high eddy activity (Chelton et al., 2004) suggesting the ocean has the potential to control wind speeds on such scales (Small et al., 2008). The differential heating of the atmosphere over oceanic fronts with strong temperature gradients invokes adjustments in the atmospheric circulation (Hsu, 1984; Wallace et al., 1989). The effects of intense sea surface temperature (SST) gradients onto the atmosphere can be seen all the way up to the tropopause (Minobe et al., 2008) and SST variability in western boundary currents impact climate variability (Larson et al., 2024). Oceanic fronts, and the associated temperature gradients, are more sharply defined in eddying models which is why at least eddy-present models, together with a sufficiently high atmospheric resolution, are required to capture these mechanisms (Tsartsali et al., 2022). Additionally, the ocean mesoscale affects large scale decadal variability of upper ocean heat content with possible effects on the atmosphere on a surprisingly long time scale (Constantinou and Hogg, 2021).

As detailed above, there are many reasons why an enhanced ocean lateral grid resolution will improve a model's ability to simulate climate variability and change. However, enhancing the ocean grid resolution does not always improve the simulation and some aspects even worsen with limited agreement between models (Haarsma et al., 2016; Chassignet et al., 2020; Chang et al., 2020; Jackson et al., 2023). The impact of ocean grid resolution on the simulated climate differs for each model and it is beneficial to test the resolution-dependency individually. Recent improvements in the configuration of the Australian Community Climate and Earth-System Simulator (ACCESS) climate model (ACCESS-CM2) suite allow for long climate simulations to be undertaken with an ocean at a higher lateral resolution of 0.25° that permit ocean eddies to be present across

**Table 1.** Overview of ACCESS model configurations used in this study and their submodel components, resolutions, number of processors used for each configuration (T: total, O: ocean, SI: sea ice, A: atmosphere), and computational efficiencies on the Australian National Computational Infrastructure supercomputer *gadi*. The core count in CM2-025 allows for similar runtimes across model components and minimal waiting for coupler communication, resulting in similar performance compared with CM2-1, which is slowed down by the atmosphere.

| Model | Ocean component | Atmosphere component | Cores T (O, SI, A) | Performance | Study |
|---|---|---|---|---|---|
| CM2-025 | MOM5, 0.25° | UM10.6 GA7.1, N96 (1.25°×1.875°) | 1152 (618, 96, 432) | 4.3 yr d$^{-1}$ | This study |
| CM2-1 | MOM5, 1° | UM10.6 GA7.1, N96 (1.25°×1.875°) | 672 (80, 16, 576) | 5.5 yr d$^{-1}$ | Bi et al. (2020) |
| OM2-025 | MOM5, 0.25° | JRA55-do v1.3 reanalysis | 1824 (1456, 361, 1) | 12.2 yr d$^{-1}$ | Kiss et al. (2020) |
| OM2-1 | MOM5, 1° | JRA55-do v1.3 reanalysis | 288 (216, 24, 1) | 95 yr d$^{-1}$ | Kiss et al. (2020) |

much of the globe. This paper investigates the extent to which ocean grid resolution improves intrinsic ocean variability, i.e. the variability induced by ocean eddies, and thereby modifies the simulated mean state and large-scale decadal climate variability in the ACCESS-CM2 model. We do this by comparing coupled (atmosphere-land-ocean-sea ice) and uncoupled (ocean-sea ice) models of the ACCESS family at 1° and 0.25° lateral resolutions of the ocean-sea ice component (with the standard N96 atmosphere resolution). The comparison with the uncoupled counterparts allow for insights whether the differences between the simulated climates predominantly emerge from the lateral grid resolution or the forced versus coupled atmosphere. The analysis focuses on known biases in ACCESS-CM2 (e.g., surface temperature and salinity, sea ice) and aspects of the climate system that we expect to respond to enhanced ocean grid resolution (e.g., ocean horizontal and overturning circulation, ocean-atmosphere interaction).

## 2 The ACCESS model

### 2.1 Model suite

ACCESS is the Australian climate modelling framework that includes atmosphere, ocean-sea ice, coupled and Earth system model configurations. This paper introduces a newly developed version of the existing ACCESS-CM2 (Bi et al., 2020) with a higher resolution ocean-sea ice component of 0.25° (CM2-025). We evaluate CM2-025 against observations, against the previous 1° coarse-resolution version of the model (CM2-1), and against two forced ACCESS ocean-sea ice configurations (ACCESS-OM2, Kiss et al., 2020) on the same horizontal grids (OM2-025 and OM2-1). An overview of the simulations analysed in this study is given in Table 1.

ACCESS-CM2 (CM2-1) constitutes one of two ACCESS model contributions to CMIP6 with the other model being the full Earth System Model ACCESS-ESM1.5 (Ziehn et al., 2020) which includes an interactive carbon cycle. A detailed description of CM2-1 is given in Bi et al. (2020), here we only report on the main features as well as the differences of CM2-025 compared with CM2-1. The atmospheric submodel is the Met Office Unified Model (UM) version 10.6 GA7.1 (Walters et al., 2019) on a N96 grid of 1.25° latitude by 1.875° longitude with 85 vertical levels. The land surface submodel is CABLE version 2.5

(Kowalczyk et al., 2006) which is a module of the atmospheric model and not independently coupled. CABLE has up to 13 surface tile types whose distribution in a grid cell is given by a prescribed time-invariant tile fraction. The ocean submodel is based on MOM version 5.1 (Griffies, 2012) and the sea ice submodel is CICE version 5.1.2 (Hunke et al., 2015), with the same horizontal resolution as the ocean model: 1° for CM2-1 and 0.25° for CM2-025. Both configurations have 50 layers in the ocean using a $z^*$ vertical coordinate system and five ice thickness categories (plus open water).

The model components are coupled three hourly using the OASIS-MCT coupler (Valcke et al., 2015; Craig et al., 2017) with an ice-ocean time step of 30 min in CM2-1 and 20 min in CM2-025. The remapping between the atmosphere and ocean-sea ice components uses first-order conservative interpolation for the flux fields and bilinear interpolation for the wind stress in CM2-1. CM2-025 uses patch interpolation for the wind stress to reduce the imprint of the the coarser atmospheric when regridded on the finer grid. The bathymetry for CM2-1 uses a legacy dataset (Bi et al., 2013) while OM2-1 and OM2-025 use a modified version of the GFDL-CM2.5 bathymetry (Griffies et al., 2015; Kiss et al., 2020). CM2-025 uses a new topography created from GEBCO_2014 v20150318 30 arc-second topography (GEBCO, 2014) which is revised in specific locations (key straits) by copying depth values from the OM2-025 bathymetry. Additional key locations also required revision including a change of the land mask around the Antarctic continental shelf to avoid model crashes during local storm events. Storms can occasionally grow stronger in the coupled system compared with the prescribed atmosphere in the OM2 models (see Section 2.2), hence it is not an issue in OM2-025. Other revisions involve the deepening of the gateways of the Mediterranean and Baltic Seas which otherwise salinify or freshen in the coupled model due to limited exchange between the respective marginal sea and the open ocean. The grid cell depth was chosen primarily to improve representation of water mass exchange, rather than to enhance overall realism. Surface salinity restoring masks the issue in the ACCESS-OM2 configurations. The sea surface salinity in ACCESS-OM2 is globally relaxed toward the World Ocean Atlas 2013 v2 (WOA13, Zweng et al., 2013, same as used for initial conditions, see Section 2.2) through a salt flux, with the strength determined by a piston velocity of 33 m per 300 d (Kiss et al., 2020). The restoring salt flux is derived from the difference between surface salinity in the model and in WOA13 and kept below ±0.5 psu to prevent excessively large fluxes.

The primary sub-gridscale parameterisations in ACCESS-CM2 and ACCESS-OM2 are similar at each resolution, including the submesoscale parameterisation (Fox-Kemper and Ferrari, 2008), biharmonic isotropic Smagorinsky horizontal friction (Griffies and Hallberg, 2000) and KPP vertical diffusivity (Large et al., 1994). The background diffusivity is set to zero at 0.25°, due to excessive numerical mixing, while the Jochum (2009) latitudinally-dependent, depth-independent background vertical tracer diffusivity scheme is used at 1°. (Details of the implementation of these parameterisations are provided by Kiss et al. (2020).) The parameterisation of mesoscale eddies is applied more judiciously, to account for the partial resolution of mesoscale features at 0.25°. The Redi diffusivity for lateral isopycnal mixing is set to a constant $600\,\mathrm{m^2 s^{-1}}$ at 1° resolution, but is scaled by the grid resolution relative to either the first baroclinic Rossby radius, or the equatorial Rossby radius for latitudes between ±5°N, with a maximum diffusivity of $200\,\mathrm{m^2 s^{-1}}$. The GM diffusivity is depth-independent but varies laterally, scaled by a combination of an inverse Eady growth rate timescale and a length scale squared (50 km at 1° and 20 km at 0.25°), as well as the same grid scaling used in the calculation of Redi diffusivity. There is a maximum diffusivity of $600\,\mathrm{m^2 s^{-1}}$ at 1°

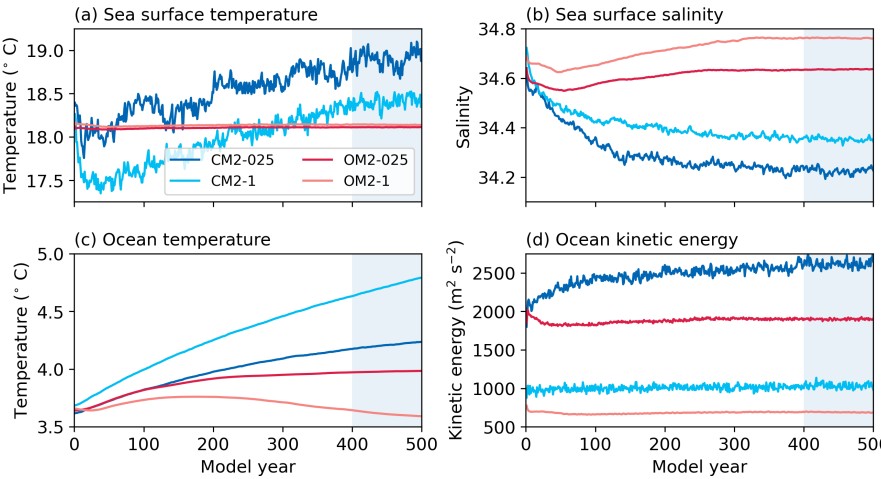

**Figure 1.** Time series of annually and globally averaged quantities for CM2-025 (dark blue), CM2-1 (light blue), OM2-025 (dark red), and OM2-1 (light red). (a) Sea surface temperature, (b) sea surface salinity, (c) ocean temperature, and (d) ocean kinetic energy. Blue shading indicates the time period used for time-mean analyses in the remainder of the manuscript.

resolution and 200 m$^2$s$^{-1}$ at 0.25° resolution. The reduced eddy parameterisation at 0.25° has been tuned in ocean-sea ice simulations with a biogeochemical model to optimise transport of tracers by the ocean's overturning circulation.

## 2.2 Model drift

All simulations are integrated over 500 years. Both CM2-1 and CM2-025 use present-day forcing representing an average of the 1985-2014 (nominally year 2000) atmospheric conditions. Using present-day forcing allows for comparison of the model output to observations which are lacking the required spatial coverage for the pre-industrial climate which is often applied for climate model control simulations. CM2-1 is initialised from the 950 year-long pre-industrial CM2-1 simulation documented in Bi et al. (2020). CM2-025's atmospheric component is also initialised from the 950 year-long pre-industrial CM2-1 simulation while the ocean component in CM2-025 is initialised from observations. The initial conditions for the ocean-sea ice component are the same as for the forced OM2 configurations: The OM2 (and CM2-025) configurations are initialised from rest with zero sea level and 1955-2012 average temperature and salinity from the WOA13 (Locarnini et al., 2013; Zweng et al., 2013). The initial sea ice field is 2.5 m poleward of 70° N and 60° S with the added condition that the SST is less than 1° C above freezing point. OM2 is forced three hourly with a repeat year forcing (May 1990 to April 1991) representative for a time period when the major climate modes of variability are neutral (Stewart et al., 2020) and that is based on the JRA55-do v1.3 atmospheric reanalysis product (Tsujino et al., 2018).

The analysis focuses on the last 100 years (model years 400-499, blue shading in Fig. 1), a period long enough to largely average out imprints of climate modes of variability present in the CM2 models but not in OM2. After the 400 year spin-up period, the simulated climate systems have largely equilibrated. The time series of globally averaged surface and interior ocean

temperatures in both CM2 simulations (Fig. 1a and c), however, indicate continuous warming. The surface and interior ocean warming (through ocean heat uptake) is reflected in energy imbalances at the top of the atmosphere (Bi et al., 2020) which is notably reduced in CM2-025 compared with CM2-1 (compare slopes in Fig. 1c). The interior ocean temperature trend in a 90-year test simulation of CM2-025 under pre-industrial conditions is reduced to 30% compared with the present-day CM2-
025 simulation presented here, suggesting a substantial contribution of the present-day forcing to the total drift. Additionally, there is a slow adjustment of the deep ocean as the formation and spreading of deep and bottom waters act on time scales of hundreds of years (Lumpkin and Speer, 2007). The OM2 configurations also indicate a slow adjustment of the interior ocean temperature, a cooling in these cases, while the surface temperatures adjust immediately to the applied repeat-year-forcing. The higher resolution models (CM2-025 and OM2-025) improve the model drift of the deep ocean, indicative that these models
better resolve ocean dynamics, which we will discuss in more detail later. A 1000 year simulation with CM2-1 reveals that the temperature trend both at the surface and in the interior ocean continues (not shown) and while this is an important issue, we stop the CM2-025 simulation after 500 years due to computational limitations and because other key quantities like sea surface salinity and ocean kinetic energy have stabilised (Fig. 1b and d).

## 3  Mean state

This section examines the climatological mean of key climate parameters with a focus on the ocean and near-surface atmosphere. The analysis is guided by known CM2-1 biases, and aims to identify whether ocean model resolution or a forced versus coupled atmosphere has a bigger impact on the simulated climate. The coupled system evolves more freely and converges to a different state than the forced ocean-sea ice model. We expect the biases, when compared to observations, therefore to be larger in the coupled system, in particular near the surface. Table 2 provides globally averaged values which summarise the
main features. Additionally, we present spatial patterns in the following subsections which are insightful as they reveal dynamical or systematic problems of the individual models. We first present the effect of the lateral grid resolution on the surface and interior ocean, on the sea ice, we then show the imprint of the changed ocean on evaporation as an example of the effect on the lower atmosphere, and finally look at modes of variability of the coupled system.

### 3.1  Surface and interior ocean

### 3.1.1  Surface temperature and salinity biases

Both CM2 configurations have a warm global mean SST bias (Table 2, 0.7° C for CM2-025 and 0.21° C for CM2-1) with a similar spatial pattern (Fig. 2a and c, also reflected in the similar root mean squared error (RMSE) between the models and observations, see Table 2). The positive bias is dominated by a too warm Southern Ocean, warm patches in the upwelling regions of the tropical eastern Pacific and Atlantic, and a too warm Indonesian Sea. Regions of cold biased SST exist in the
mid-latitudes of the Northern Hemisphere Pacific and Atlantic as well as the high latitude Atlantic. The imbalanced geographical distribution of the warm and cold biases can largely be ascribed to the parameterisation of clouds and aerosols in the

**Table 2.** Overview of different integrated variables for observations (where available) and the four ACCESS models considered in this study. RMSE is the root mean squared error between the models and observations, SST stands for sea surface temperature, SSS stands for sea surface salinity, ACC stands for Antarctic Circumpolar Current, and AMOC stands for Atlantic Meridional Overturning Circulation.

| Diagnostic | Observations | CM2-025 | CM2-1 | OM2-025 | OM2-1 |
|---|---|---|---|---|---|
| Global mean, RMSE | | | | | |
| - SST ($^\circ$ C) | $18.2^a$ | 18.9, 1.2 | 18.41, 1.03 | 18.11, 0.62 | 18.14, 0.64 |
| - SSS (psu) | 34.33 | 34.23, 0.87 | 34.35, 1.22 | 34.64, 0.5 | 34.76, 0.38 |
| Transport through straits (Sv): | | | | | |
| - Drake Passage | $173.3\pm10.7^b$ | 210.65 | 182.71 | 146.68 | 152.38 |
| - Indonesian Throughflow, Total | $-15^c$ | -8.63 | -6.72 | -7.75 | -6.54 |
|     Lombok | $-2.6^b$ | -3.49 | -1.19 | -3.47 | -0.96 |
|     Ombai | $-4.9^b$ | -0.04 | 0.29 | -0.21 | 0.75 |
|     Timor | $-7.5^b$ | -5.1 | -5.81 | -4.08 | -6.33 |
| Total kinetic energy, ocean (PJ) | - | 2617 | 1034 | 1899 | 690 |
| Mixed layer depth, max (m): | | | | | |
| - Nordic Sea, Irminger Sea | $1107, 663^d$ | 1815, 1174 | 1752, 1906 | 1541, 2019 | 1523, 2057 |
| - Labrador Sea | 1229 | 1197 | 326 | 3693 | 2957 |
| - Kuroshio, Gulf Stream | 176, 272 | 352, 510 | 359, 593 | 1306, 559 | 589, 634 |
| - ACC | 569 | 810 | 676 | 959 | 989 |
| - Weddell Sea, Ross Sea | 561, 395 | 3727, 4243 | 3074, 3891 | 4862, 4371 | 4923, 4429 |
| Overturning circulation (Sv): | | | | | |
| - Abyssal overturning at 40$^\circ$ S | - | 13.51 | 5.04 | 8.7 | 11.49 |
| - AMOC at 26$^\circ$ N | $17.2\pm0.9^e$ | 16.63 | 16.28 | 15.1 | 10.67 |
| Sea ice extent (max, min in $10^6$ km$^3$): | | | | | |
| - Antarctic | $19.45, 3.66^f$ | 16.41, 0.4 | 18.13, 1.19 | 18.9, 1.64 | 18.46, 1.22 |
| - Arctic | $14.43, 7.02^f$ | 17.28, 6.62 | 18.04, 7.62 | 15.65, 6.52 | 15.95, 6.42 |

[a] WOA13, [b] Donohue et al. (2016), [c] Sprintall et al. (2009),

[d] De Boyer Montégut (2023), [e] McCarthy et al. (2015) years 2004-2012 , [f] NSIDC.

atmospheric model (Bi et al., 2020), explaining the similarity of the spatial patterns in CM2-025 and CM2-1. In particular, the warm bias in the Southern Ocean is predominantly due to too little cloud coverage and therefore too much incoming shortwave radiation, a common issue amongst climate models (Hyder et al., 2018). The ocean-only OM2-025 configuration displays a

number of Southern Ocean SST bias features that are also present in the CM2-025 run, albeit with smaller magnitude, suggesting that ocean physics errors are also a contributor to these biases (see end of paragraph). The response is amplified in CM2-025 in the Southern Ocean (mostly in the Pacific sector) while it is reduced in the Northern Hemisphere Pacific leading to the globally averaged larger bias in CM2-025 compared with CM2-1. The magnitude of the warm biases off the west coasts of South America and Africa are similar for both model resolutions. These regions are characterised by wind-driven

upwelling where surface waters are transported westward and replaced by cold waters from below, a processes that is not well represented in either of the model configurations. It is likely that the simulated winds are the main cause of the problem as the atmospheric model is the same between CM2-1 and CM2-025 and therefore explains why the higher ocean resolution does not considerably aid to reduce the bias. The biases of the forced simulations have a similar pattern but are noticeably muted (Table 2, -0.09° C for OM2-025 and -0.06° C for OM2-1, with an almost halved RMSE) as the ocean surface is directly exposed to

the fluxes based on the reanalysis product and is therefore close to observations. Dynamically important regions such as the western boundary currents and to a lesser extent the tropical Pacific, the upwelling regions off the coast of Africa as well as the Southern Ocean also emerge in the SST anomalies (Fig. 2b), highlighting the inability of the ocean model to adequately represent these dynamical features even at 0.25° nominal grid resolution.

The simulated global mean sea surface salinity (SSS) in CM2-1 matches well the observed estimates (0.02, Table 2), while

CM2-025 is biased fresh (-0.1). Similar to SST, however, the global mean hides large spatial patterns of sizeable magnitude revealing problems of both models to adequately simulate various dynamical features (Fig. 3, RMSE in Table 2). Most pronounced is the fresh anomaly in the western tropical Pacific. The bias occurs because the South Pacific Convergence Zone is not well represented with a too zonal cross-basin slope originating from the warm SST bias in the eastern upwelling region and which results in too much precipitation in the west (Folland et al., 2002; Bi et al., 2020). The Bay of Bengal shows a positive

bias which is related to a negative rainfall bias in the atmospheric model component (Bi et al., 2020). Large biases also occur in the Arctic ocean where sea ice has a large imprint on the SSS and river runoff from the land model is known to have errors (Bi et al., 2020). Similar positive salinity biases in the Canadian Basin are ubiquitous across CMIP6 models (Khosravi et al., 2022). The models poorly represent this part of the globe, however, note that observations of the often ice-covered ocean that we use in this comparison might be flawed (Zweng et al., 2013). The forced ocean-only models have a modest sea surface salinity

restoring to the World Ocean Atlas 2013 v2 (the restoring flux is kept below ±0.5 psu). While they have a larger global mean SSS bias compared to the fully coupled models, this number is dominated by the Arctic Ocean and the RMSE is therefore much reduced (Table 2, Fig. 3b). The SSS anomaly in OM2-025 has elsewhere a much smaller magnitude. The large values in the Arctic, where the ocean is covered by sea ice over much of the year, indicates the prescribed surface salinity restoring in OM2 struggles to rectify issues emerging from ocean-sea ice interaction. Other challenging regions to simulate are the marginal seas

where there is limited exchange with the open ocean resulting in a delicate balance between exchange, precipitation, evaporation and river runoff. In particular, the salinity in CM2-025 in the Baltic and Mediterranean Seas did not stabilise when

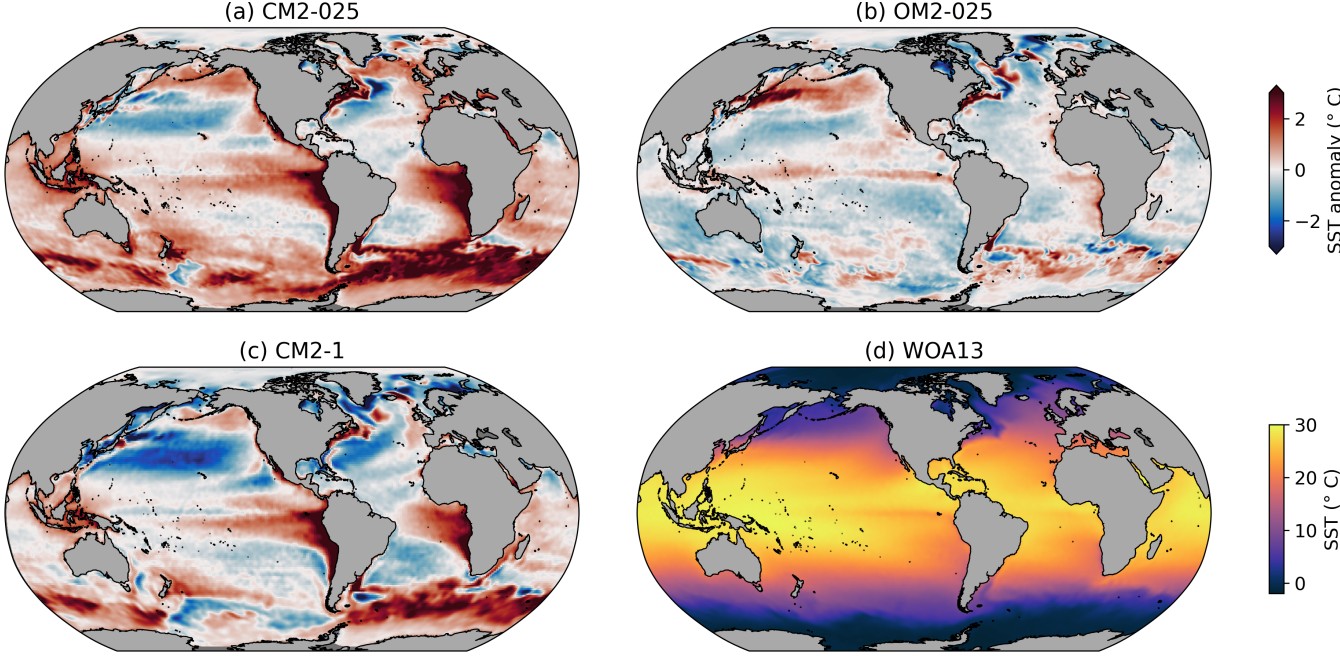

**Figure 2.** Time-mean sea surface temperature (SST) biases in (a) CM2-025, (b) OM2-025, and (c) CM2-1 relative to (d) observations. Observations are a 1955-2012 average based on World Ocean Atlas 2013. Missing data is shown in dark grey, land in light grey.

developing CM2-025. The CM2-025 topography has therefore been deepened to ensure sufficient exchange between the open ocean and the marginal sea. While this approach avoids a drift in salinity, the salinity biases remain large (Figure 3a). Other regions such as the Black Sea or the Persian Gulf could potentially benefit from similar changes to the bathymetry product but these were not considered necessary. The issue does not emerge in the forced models due to large surface salinity restoring in these regions (not shown).

### 3.1.2 Horizontal circulation

The ACC is the strongest ocean current that connects all main ocean basins. Its transport through Drake Passage is an informative integrated metric for the representation of the ocean circulation mean state. The strength of the ACC is set by the meridional density gradient which in turn depends on multiple factors including wind stress forcing at the ocean surface, temperature and salinity properties of the surface and interior ocean, eddy activity, and sea ice formation and melt. The representation of all these aspects impact the simulated ACC and vice versa.

Both ACCESS-CM2 models have a stronger ACC transport than the observational estimate of 173 Sv (Donohue et al., 2016), although CM2-1's 183 Sv is within the error range of the observations (Table 2). The larger transport of 211 Sv in CM2-025along with a larger SST bias in the Southern Ocean which increases the vertical stratification, is consistent with a stronger thermal wind contribution to zonal velocity (Sect. 3.1.1, Fig. 2). These biases offset any benefits of partially resolving eddies in

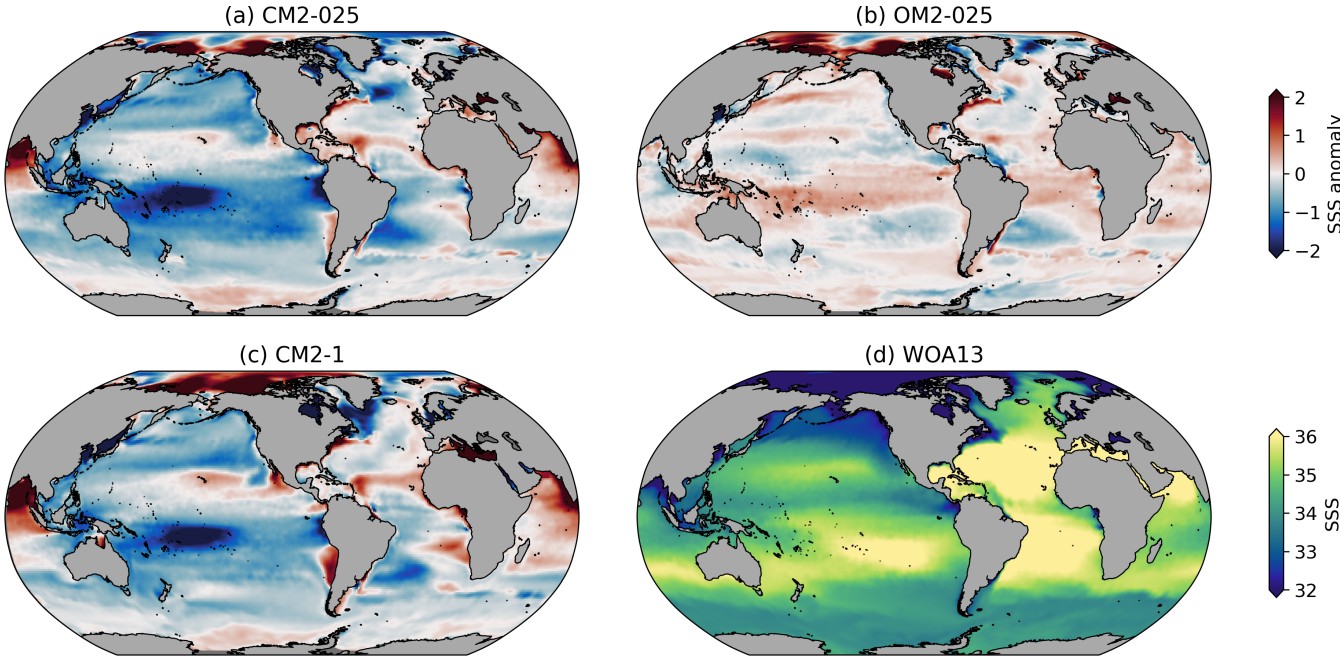

**Figure 3.** Time-mean sea surface salinity (SSS) biases in (a) CM2-025, (b) OM2-025, and (c) CM2-1 relative to (d) observations. Observations are a 1955-2012 average based on World Ocean Atlas 2013. Missing data is shown in dark grey, land in light grey.

CM2-025 which act to flatten the slope of the isopycnals (surfaces of constant density) and reduce the ACC strength (Munday et al., 2013). The forced ACCESS-OM2 models both have a weaker transport through Drake Passage which is common for this type of model (Tsujino et al., 2020) and most CMIP6 models (Beadling et al., 2020). The higher resolution model OM2-025 has with 147 Sv a weaker transport than its coarser counterpart OM2-1 with 153 Sv (Table 2). For both model configurations, the coarser resolution models compare better with the observation estimate in terms of the integrated transport through Drake Passage. However, their simulated ACC is a single broad jet. The ACC is in reality made up of multiple strong and narrow jets and CM2-025 and OM2-025 much better capture this feature (not shown). The fact that both higher resolution models exhibit the same behaviour confirm the requirement of a higher grid resolution and lower eddy diffusivity to better represent the eddying structure of the ACC, and associated vertical momentum transfer (Ward and Hogg, 2011), even if the integrated transport deviates more from the observational estimate.

The Indonesian Throughflow is another useful integrated metric for ocean circulation as it is the only tropical inter-basin exchange between the Pacific and Indian Oceans. A sea level gradient between the two ocean basins sets up a transport from the Pacific into the Indian Ocean of 15 Sv (Sprintall et al., 2009). The transport occurs through three narrow straits (Lombok Strait, Ombai Strait and Timor Passage) and coarse resolution models typically struggle to represent the circulation in this region (Tsujino et al., 2020). It comes at no surprise that all four ACCESS configurations considered in this study underestimate the total estimated transport by a large margin (Table 2). The transport is somewhat larger in the eddy-permitting configurations

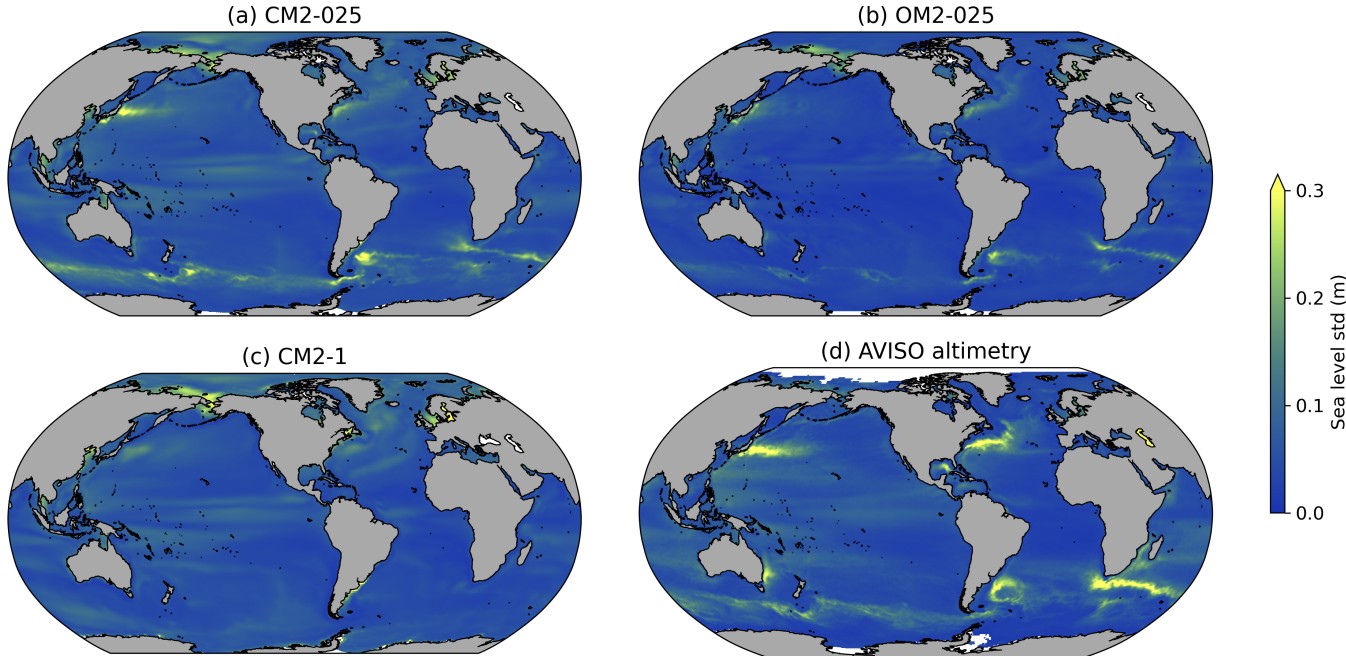

**Figure 4.** Maps of sea level anomaly standard deviation for (a) CM2-025, (b) OM2-025, (c) CM2-1, and (d) observational estimate from AVISO altimetry.

(8.6 Sv in CM2-025 and 7.8 Sv in OM2-025 compared with 6.7 Sv in CM2-1 and 6.5 Sv in OM2-1) but the resolution is still not sufficient to adequately resolve the total transport. Kiss et al. (2020) identify three main reasons for the weaker transport
in ACCESS-OM2 which we expect to hold for ACCESS-CM2. First, the simulated sea level gradient between the two ocean basins is too weak. Second, the narrow Lombok and Ombai Straits are only represented by a single grid cell in the 0.25° and 1° configurations. Lastly, the models struggle to correctly partition the transport between the three passages with too much transport through the first Strait (Lombok Strait) and too little through the second (Ombai Strait) and third (Timor Passage) pathways (Table 2).

We next consider the spatial variability of sea level anomalies as a measure of surface mesoscale variability (Fig. 4). The standard deviation of monthly sea level anomaly serves as a good proxy for eddy activity and provides the possibility for comparison with an observational product. We make use of the 1993-2014 sea level estimate from AVISO SSALTO/DUACS altimetry (http://marine.copernicus.eu and https://www.aviso.altimetry.fr). The western boundary currents are the regions of highest eddy activity (Fig. 4d). Enhanced eddy activity also develops in the Southern Ocean along the ACC. Broader regions
of modest eddy activity exist in the lower latitudes of the Pacific and Indian Ocean which are connected to climate modes of variability, that is El Nino-Southern Oscillation (ENSO) in the Pacific and the Indian Ocean Dipole in the Indian Ocean. CM2-025 best captures both the spatial pattern and the amplitude of the observed mesoscale variability although it still underestimates the observational estimates (Fig. 4a). OM2-025 also captures the spatial pattern but with much lower amplitudes. In particular

the enhanced mesoscale variability in the low latitudes is much reduced, indicative that the modes of variability involving the atmosphere are not well represented in the forced ocean model. CM2-1 is able to capture the variability of the low latitudes but barely exhibits any enhanced variability in the eddy-active western boundary currents and ACC. The comparison of the sea level variability shows clearly that the higher grid resolution much improves the representation of mesoscale variability as one would expect when entering the eddy-permitting regime. The higher mesoscale activity also leads to a higher total ocean kinetic energy. CM2-025 (2617 PJ) has the highest kinetic energy followed by OM2-025 (1899 PJ), then CM2-1 (1034 PJ) and OM2-1 (690 PJ) has the lowest total kinetic energy (Table 2). The comparison shows that the grid resolution is most important factor but the coupled models, which can evolve more freely, are more energetic than the forced model configurations.

### 3.1.3 Mixed layer depth

Momentum and buoyancy fluxes between the ocean and the atmosphere (or sea ice) alter the ocean over the entire surface mixed layer. The surface mixed layer, which is a relatively homogeneous layer characterised by turbulent mixing and low stratification, links in turn the surface to the interior ocean via water-mass formation. The surface mixed layer is therefore an important part of the climate system and a key model diagnostic. The mixed layer depth (MLD) is very sensitive to surface fluxes and internal ocean dynamics due to its low stratification and varies substantially in space and time (Monterey and Levitus, 1997; de Boyer Montégut et al., 2004; Holte et al., 2017), leading to large model biases in both coupled and uncoupled models (Tsujino et al., 2020; Heuzé, 2021).

Different approaches exist to define the MLD which lead to uncertainties in the observational estimates in the order of 10 % (Holte et al., 2017). We compare the model output to observations provided by the Sea Scientific Open Data Publication (https://www.seanoe.org/data/00806/91774/) which uses a density threshold criterion of 0.03 kg m$^{-3}$ and a 10 m reference depth (De Boyer Montégut, 2023). The MLD in the model is the depth at which the buoyancy exceeds the surface buoyancy by more than the default buoyancy criterion (0.0003 m s$^{-2}$), following a recommendation for ocean models (Griffies et al., 2016). The buoyancy-based definition used in the models approximates the density-based definition used in the observational product to within 5%. For the purpose of this model validation, where we focus on the bias of the maximum MLD, the errors introduced due to the different choice of MLD definition is secondary as the models present large deep biases and observations tend to overestimate rather than underestimate MLD (de Boyer Montégut et al., 2004). Figure 5 shows the maximum MLD during the year based on a monthly climatology for a given grid point (biases relative to observations are shown for the models). Figure 6 shows the seasonal evolution of the maximum monthly climatological MLD for selected regions. The MLD varies substantially over the globe from tens of metres in the tropics to a few hundred metres in the high latitudes where turbulent mixing from strong winds and buoyancy loss reduces the surface density resulting in deep winter mixed layers (Fig. 5d). A realistic representation of winter MLD in the high latitudes is critical for deep and bottom water formation with consequences for the ocean's overturning circulation (Sect. 3.1.4) and carbon uptake. Both the coupled and uncoupled models overestimate the maximum MLD in regions of deep MLD while summertime values are comparable (Fig. 5a-c, Fig. 6a-c, g-h).

In the Southern Hemisphere, the ACCESS models simulate very deep MLD in the Ross and Weddell Seas where they overestimate deep convection in wintertime open ocean polynyas (Fig. 5, Fig. 6 and Table 2). Open ocean polynyas have

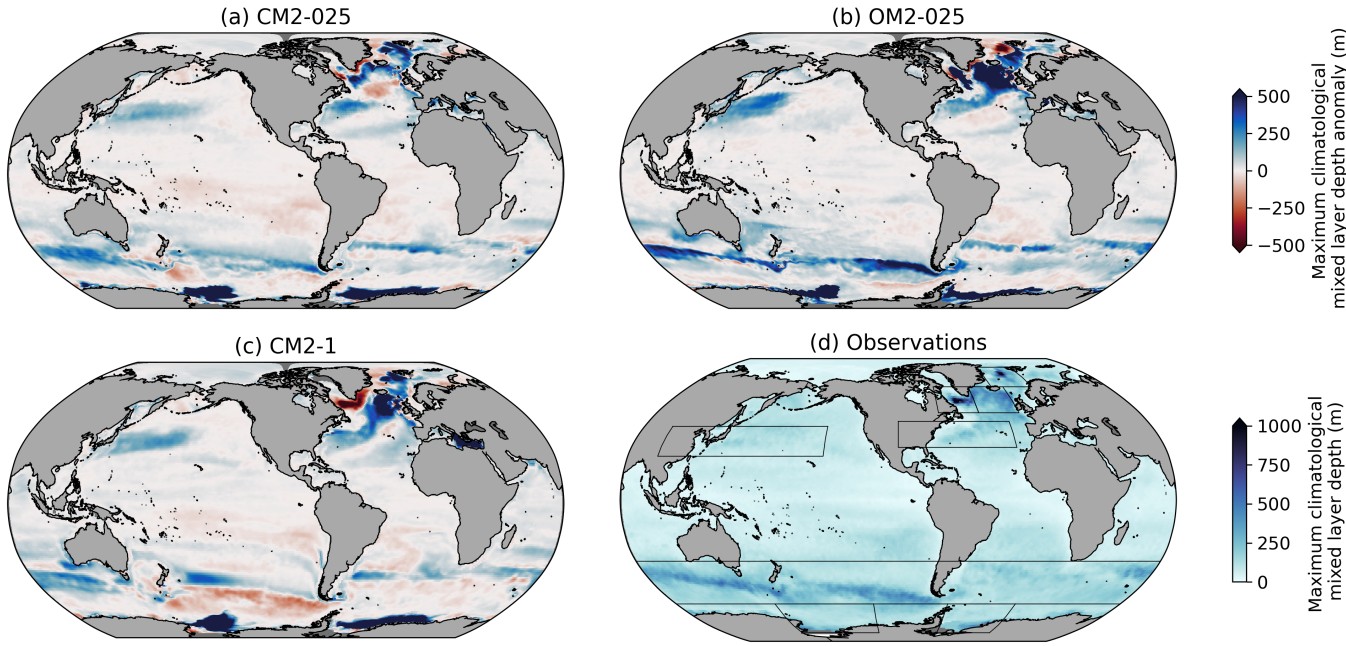

**Figure 5.** Maximum monthly climatological mixed layer depth biases for a given location in (a) CM2-025, (b) OM2-025, and (c) CM2-1 relative to (d) observational derived mixed layer depths for 1979-2021 from De Boyer Montégut (2023). The black boxes indicate the regions for which maximum mixed layer depths are reported in Table 2 and Figure 6 Missing data is shown in dark grey, land in light grey.

been reported twice in the Weddell Sea for the observational period (Gordon, 1978; Carsey, 1980; Campbell et al., 2019), but are a common feature in most climate models at various grid resolutions, including ACCESS-CM2 (Heuzé, 2021). The
misrepresentation of deep convection affects the formation of Antarctic Bottom Water (AABW) which, in reality, is formed via Dense Shelf Water (DSW) formation in coastal polynyas on the Antarctic continental shelf and mixing with adjacent waters when flowing down the continental slope (Gill, 1973; Jacobs et al., 1970; Silvano et al., 2020). OM2-025 also exhibits deep convection in open ocean polynyas in the Ross and Weddell Seas (Fig. 5b, Fig. 6g-h). Only a configuration of ACCESS-OM2 with a much higher nominal grid resolution of 0.1° (not included in this study) is able to realistically form AABW (Kiss et al.,
2020; Moorman et al., 2020). Both OM2 models exhibit a deeper maximum wintertime MLD between June and November. The deepening occurs within about a month and then remains relatively constant through winter until November. In contrast, the CM2 models display a more gradual (and realistic) deepening, reaching their maximum in October. The faster MLD deepening in the OM2 models, which is not limited to the MLD evolution in the Weddell and Ross Seas (Fig. 6), likely results from the infinite heat capacity of the prescribed atmosphere, which enables rapid ocean heat loss. The MLD evolution is slower in the
CM2 models as the atmosphere must adjust to the heat flux from the ocean.

Similar to the Southern Ocean, the too deep MLD in the North Atlantic (Fig. 5 and Table 2) also indicates issues with the representation of deep convection in the Northern Hemisphere and the formation of North Atlantic Deep Water (NADW).

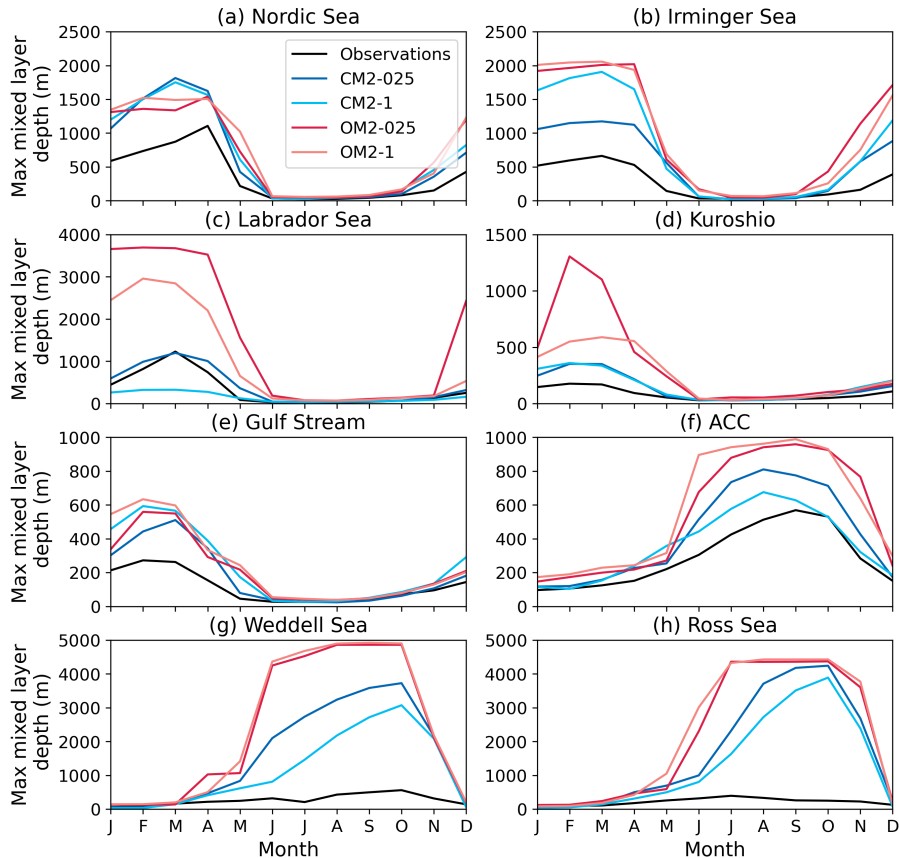

**Figure 6.** Seasonal evolution of maximum monthly climatological mixed layer depth in different regions for observations (De Boyer Montégut, 2023, black), CM2-025 (dark blue), CM2-1 (light blue), OM2-025 (dark red), and OM2-1 (light red). The different regions are shown as black boxes in Figure 5.

Observations indicate the main convection areas to be in the Nordic Seas (north of Iceland) and in the Labrador Sea. The models, however, also convect in an unrealistically large area in the Irminger Sea (south of Iceland). Deep convection is improved in the CM2-025 model compared to all other models tested here. CM2-025 does not convect as much in the region south of Iceland and is able to convect in the Labrador Sea, which the coarser CM2-1 model does not (Fig.5c and Fig.6c). Note, however, that deep convection in the Labrador Sea in CM2-025 is subject to a multidecadal variability which is discussed in Section 4.2.

The models also exhibit deeper maximum MLD along the ACC, the Gulf Stream, and the Kuroshio compared with the observational estimates (Fig. 6, Table 2). For the ACC, the deeper mixed layers are associated with Subantarctic Mode Water formation. For the Gulf Stream and the Kuroshio, the deeper mixed layers are associated with $18°$ C water formation. The observations show rather modest maximum MLD in these regions that are overestimates by all models. The CM2 models exhibit slight improvements both for the peak wintertime MLD value and for the seasonal evolution of MLD.

The comparison between CM2-025 and OM2-025 suggests deficiencies in the ocean model component. All configurations overestimate the maximum MLD which suggests the atmospheric forcing is less likely to be primary source for the biases. Shifting to an eddy-present model grid does not lead to uniform improvements across the globe and even worsens the simulation in some regions (e.g. along the ACC). Previous comparison studies of forced ocean-sea ice models also find a large model spread with little dependence on the applied atmospheric forcing nor on the eddy-present grid resolution (Tsujino et al., 2020; Treguier et al., 2023). Improving the representation of MLD might therefore require a much higher grid resolution in both the horizontal and vertical (Kiss et al., 2020) as well as improvements to the vertical mixing schemes and mesoscale and submesoscale parametrisations which help to set the interior stratification by re-stratifying the water column in winter.

### 3.1.4 Overturning circulation and interior ocean biases

The combined vertical and horizontal movement of water in the global ocean is encapsulated by the meridional overturning circulation. The zonally integrated overturning streamfunction has two interhemispheric cells that transport heat, salt, carbon and nutrients. The upper cell (green in Fig. 7), also known as the AMOC, is characterised by deep water formation in the polar North Atlantic (i.e. NADW) and upwelling in the Southern Ocean. In the Southern Ocean, the water moves upward along sloped isopycnals towards the surface, is transported northward by winds, and is transformed into lighter mode and intermediate waters (Marshall and Speer, 2012). Further south, a portion of the upwelled deep water from the upper cell is not transported northward but transformed into bottom water (i.e. AABW) as part of the abyssal cell (blue in Fig. 7). Diapycnal mixing, i.e. across isopycnals, near the bottom transforms the bottom water into deep water when the water moves northward and closes the abyssal cell. The abyssal cell appears small in density space as shown in Fig. 7 but does comprise a large volume of water in the ocean. Additional to the two main overturning cells, two shallow subtropical cells exist which are strongest in the Indo-Pacific and only very weak in the Atlantic and therefore hardly emerge in Fig. 7 which displays the Atlantic overturning circulation north of 4° S.

The consideration of the overturning circulation provides an insight into the ability of the models to simulate various aspects of the ocean dynamics and their far-field impact. For example, the mis-representation of deep convection in the North Atlantic and Southern Ocean (Sect. 3.1.3) affect deep and bottom water formation. The effect on the overturning can be seen by the differences in the strength and density range of the upper and abyssal cells as simulated by the ACCESS models (Fig. 7 and Table 2). The strength of the upper cell at 26°N can be compared to available observations (McCarthy et al., 2015) by calculating the maximum transport in the vertical (where density is the vertical coordinate for the models and depth is the vertical coordinate for the observations). The observational estimate of 17.2 Sv is underestimated in all ACCESS configurations (Table 2). Both coupled models (16.63 Sv for CM2-025 and 16.8 Sv for CM2-1) and also the eddy-permitting forced OM2-025 (15.1 Sv) are reasonably close, while the coarse OM2-1 has a much weaker transport (10.7 Sv). Depth estimates of the simulated maximum transports are obtained by converting the model overturning from density to depth coordinates. The observed peak transport occurs at a depth of 1030 m and is biased shallow in the coupled models (782 m in CM2-025, 969 m in CM2-1), matches the observed depth well in OM2-025 (1030 m) and is biased deep in OM2-1 (1173 m). All models are able to capture the inter-hemispheric character of the AMOC although this is better represented in the forced ocean models.

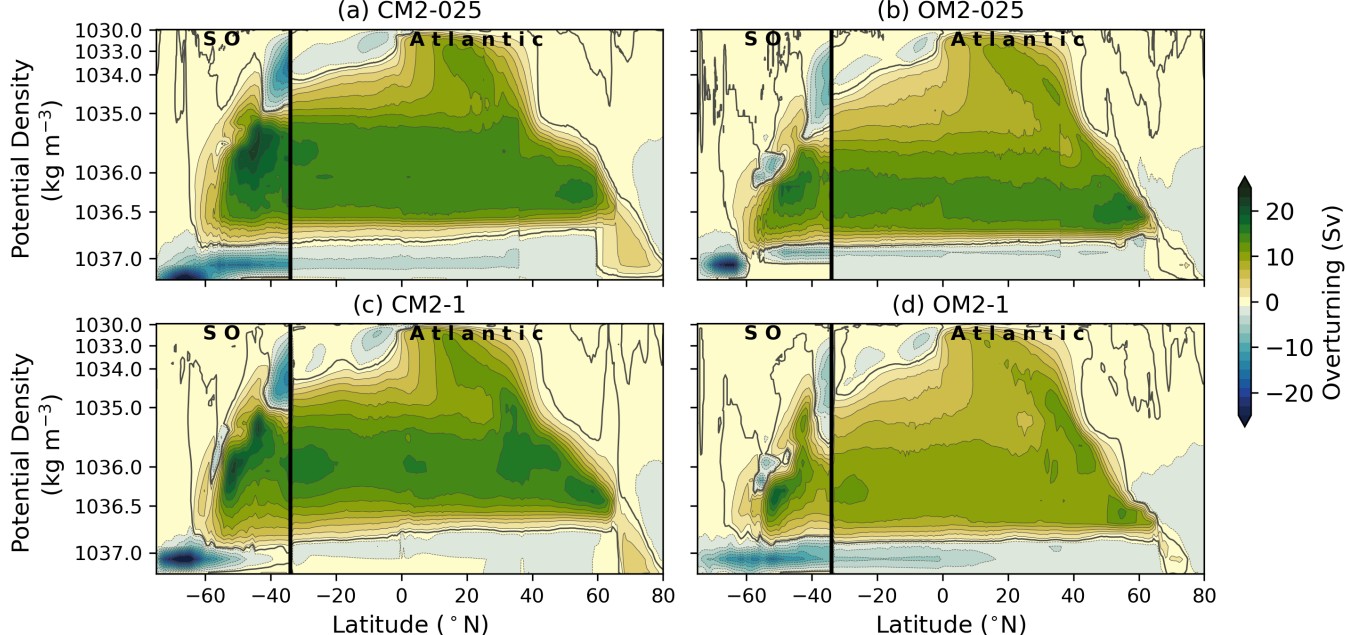

**Figure 7.** Zonally integrated meridional overturning circulation, including the transport from the eddy closure schemes, on potential density surfaces referenced to 2000 dbar for the Southern Ocean (SO) and Atlantic ocean basin, separated at $34°$ S as highlighted by the black vertical line, in (a) CM2-025, (b) OM2-025, (c) CM2-1, and (d) OM2-1. Note the non-uniform density axis. Positive (green) indicates a clockwise circulation and negative (blue) a counter-clockwise circulation. Contours are given in 2 kg m$^{-3}$ intervals and the zero contour is highlighted in bold.

The coupled models exhibit a freshening of the density class where the maximum transport occurs towards the south (between $60°$ S and $40°$ S). At the same time, the lower boundary of the AMOC cell is denser in the south compared with the Northern

Hemisphere (sloped black contour between 1036.5 kg m$^{-3}$ and 1037 kg m$^{-3}$ in Fig. 7).

    The transport of the abyssal cell is poorly constrained by observations with estimates ranging from 20 Sv at $32°$ S (Lumpkin and Speer, 2007), to 29 Sv at $30°$ S (Talley, 2013), and 50 Sv between 30-40$°$ S (Sloyan and Rintoul, 2001). The abyssal overturning cell in all models considered in this study suffer from excessive deep convection in the large open ocean polynyas in the Weddell and Ross Seas (Fig. 5). At $40°$ S, all ACCESS models are biased low with considerable differences in their

transport (13.51 Sv in CM2-025, 5 Sv in CM2-2, 8.7 Sv in OM2-025, and 11.5 Sv in OM2-1). However, the transports at $40°$ S are much reduced from the maximum transports south of $60°$ S due to the strong and unrealistic equatorward decay of the abyssal cells in the models. The large spread of transport rates at $40°$ S in the ACCESS models is therefore also an indicator of the varying meridional connectivity.

    CM2-025 forms noticeably denser AABW than observed with colder and saltier waters throughout the water column south

of $55°$ S (Fig. 8 a and b). Below 3000 m, the bias spreads to the Northern Hemisphere (Fig. 7a and Fig. 8a and b). This deep

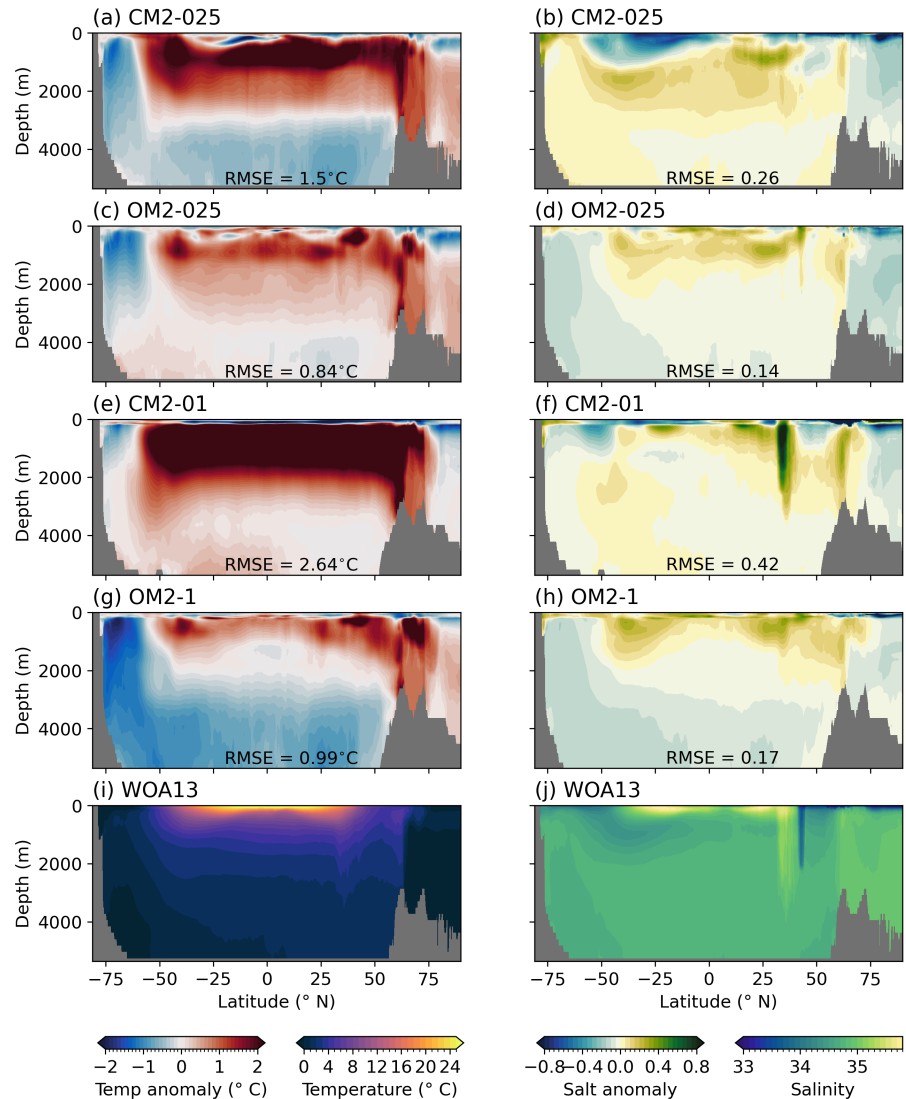

**Figure 8.** Zonally averaged temperature (left) and salinity (right) anomalies for (a-b) CM2-025, (c-d) OM2-025, (e-f) CM2-1, and (g-h) OM2-1 compared with (i-j) observations from World Ocean Atlas 13. The RMSE displayed in (a-h) is the root mean squared error between the model and observations.

temperature and salinity bias is much more pronounced in CM2-025 than in the coarser CM2-1 which exhibits a weak warm bias at depth (Fig. 8). OM2-025 has the same cold bias as CM2-025 in the far south but AABW is not as dense as the waters are fresher (Fig. 8c and d). OM2-1 has of all models considered here the largest AABW bias and simulates too cold and too fresh waters (Fig. 8g and h). The inability of the models to simulate realistic deep convection also shows in the Northern Hemisphere at around 60° N, where the models have warm and salty biases throughout the water column.

The largest interior bias in water mass properties, however, is a warm bias in the subsurface down to about 3000 m and between 55° S and 55° N. The warm bias goes along with high salinity values (fresh subsurface bias in the Southern Hemisphere for CM2-025) and exists in all models, but is more pronounced in the coupled models with some improvements in CM2-025 over CM2-1. These biases are indicative of erroneous formation and northward spread of cold and fresh Antarctic Intermediate Water. Finally, note the peak in the salinity at around 40° N in CM2-1 (Fig. 8f) due to the mis-representation of the outflow from the Mediterranean Sea. The bias is much improved in both 0.25° configurations, and in particular in CM2-025 where the bathymetry has been adjusted to allow for exchange between the open ocean and the marginal sea.

As described in the previous paragraphs, the simulated overturning circulations are approximately comparable but differences emerge for both the different model resolutions and the forced versus coupled atmosphere. The surface ocean is similar in the two coupled and the two forced configurations as the surface ocean is exposed to a similar atmosphere with similar wind stress in each case. Differences emerge predominantly in the interior ocean for denser density classes where deep convection plays an important role which varies with the ocean grid resolution. CM2-025 has a stronger abyssal cell and weaker AMOC compared with the coarser CM2-1. This is opposite in the forced configurations where OM2-025 has a weaker abyssal cell and stronger AMOC compared with OM2-1. It is common that models with a stronger abyssal cell have a weaker upper cell and vice versa. The strength of the overturning in the Southern Ocean is also linked to the simulated ACC transport as the density of AABW affects the meridional density gradient and hence the thermal wind balance. For example, the strong abyssal cell in CM2-025 goes along with a large ACC transport while OM2-025 has both the weakest abyssal cell and weakest ACC transport (Table 2).

## 3.2  Sea ice

Sea ice is at the interface between the ocean and the atmosphere and is subject to fluxes between both components. Errors in the simulated sea ice propagate into the atmosphere and the ocean; and the misrepresentation of both model subcomponents impacts vice versa the sea ice. Sea ice is therefore an important diagnostic to measure model performance. The JRA55-do reanalysis product used to force the ACCESS-OM2 models incorporates observed sea ice observations and has a strong imprint onto the simulated sea ice (Kiss et al., 2020). The forced ocean-sea ice models therefore compare reasonably well with the observed sea ice concentrations (Table 2 and Fig. 9 and 10). In contrast, the sea ice in the coupled models can evolve much more freely and is not expected to match the observations as well. There is a large intermodel spread of sea ice representation amongst CMIP6 models for both hemispheres (Shu et al., 2020; Roach et al., 2020; Notz, 2020). We assess the representation of sea ice in the ACCESS models in more detail by comparing the simulated sea ice extent to observations from the National Snow and Ice Data Centre (NSIDC,  Meier et al., 2013). We calculate the sea ice extent as the total area of grid cells with a sea ice concentration of at least 15 %.

In the Southern Hemisphere, the ACCESS-CM2 models underestimate sea ice extent throughout the year (Fig. 9a). The maximum sea ice extent in winter reaches 84 % for CM2-025 and 93 % for CM2-1 of the observed extent (Table 2). The representation of the minimum summer sea ice extent is 11 % for CM2-025 and 33 % for CM2-1, which is rather poor. The low sea ice extent in the ACCESS-CM2 models goes along with their warm SST biases in the Southern Ocean which inhibits

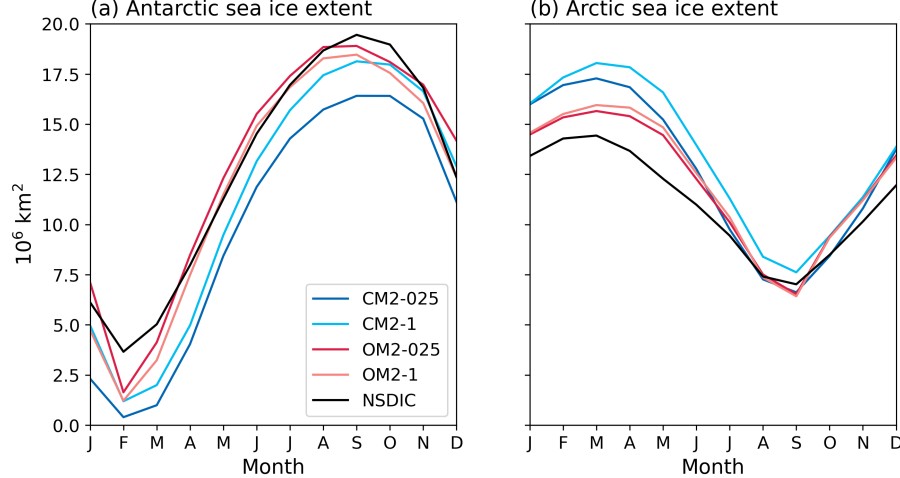

**Figure 9.** Monthly climatology of (a) Antarctic and (b) Arctic sea ice extent for observations (black), CM2-025 (dark blue), CM2-1 (light blue), OM2-025 (dark red), and OM2-1 (light red). The sea ice extent is the total area of grid cells with a sea ice concentration $\geq 15\%$. Observations are taken from the National Snow and Ice Data Centre (1979-2019).

them to retain sea ice in summer (Fig. 2). The larger Southern Ocean SST bias in CM2-025 therefore also explains why the representation of sea ice in the higher resolution model is even worse when compared with its lower resolution counterpart. The ACCESS-OM2 models capture the magnitude of the sea ice maximum well (97 % for OM2-025 and 95 % for OM2-1) although the sea ice grows faster with the maximum occurring a month earlier than in the observations. Similar to the ACCESS-CM2 models, there is too little sea ice coverage in austral summer (45 % for OM2-025 and 33 % for OM2-1).

All models have a similar spatial bias of the maximum Antarctic sea ice extent when compared with observations (Fig. 10a), i.e., they underestimate the sea ice coverage between the Weddell Sea and Prydz Bay, have a reasonable representation in East Antarctica and the Ross Sea, and overestimate the sea ice extent in West Antarctica. CM2-025 is an exception to the latter with too little sea ice extent in West Antarctica. The spatial bias of the summer minimum sea ice coverage varies between the coupled and forced models with the biggest anomalies occurring in the Weddell Sea and West Antarctica (Fig. 10c). The coupled models have too little sea ice throughout the Weddell Sea with the sea ice edge being much closer to the continent than in observations. The forced models do simulate sea ice in the west where sea ice is transported offshore in the Weddell Gyre. In West Antarctica, the sea ice is almost completely melted in the coupled models while the forced models still have a patch of sea ice which is, however, not connected to the coast unlike in observations. Note, that the ocean surface in areas of deep convection (Fig. 5) is covered by winter sea ice in all simulations, even if the concentration is lower than observed (not shown).

In the Northern Hemisphere, the maximum sea ice extent is too large in all models while the minimum sea ice extent matches the observations well (Table 2). The coupled model configurations simulate 120 % for CM2-025 and 125 % for CM2-1 of the observed maximum sea ice extent. The forced models have a very similar maximum sea ice extent to CM2-025 of 108 % for

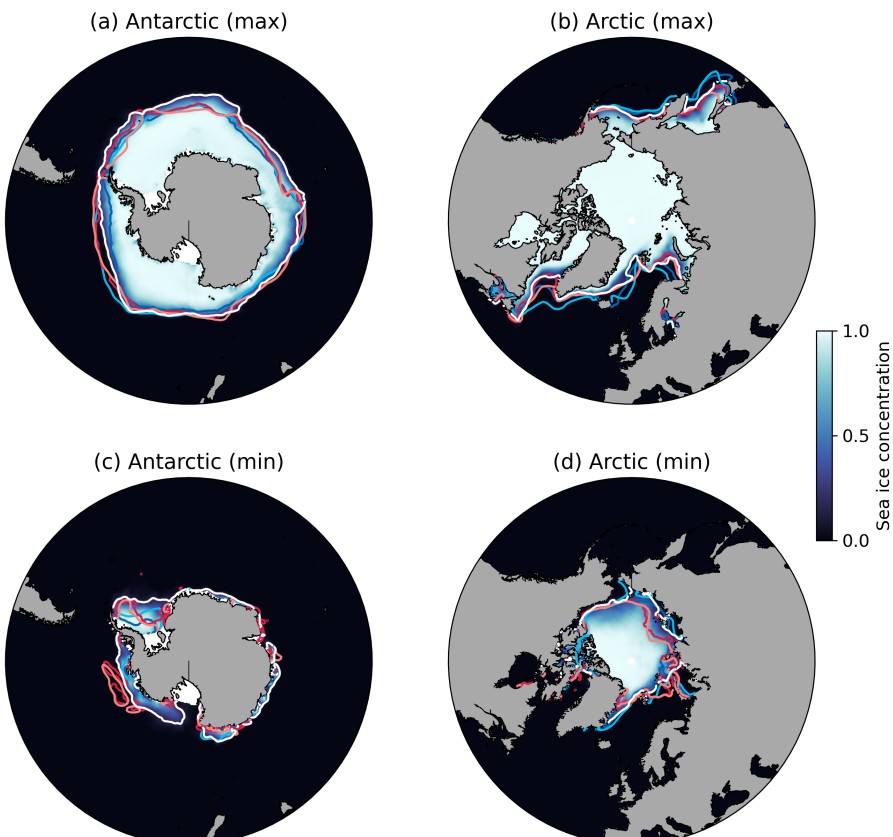

**Figure 10.** Observed maximum (top) and minimum (bottom) sea ice concentration for Antarctica (left) and the Arctic (right) from the National Snow and Ice Data Centre (1979-2019). The coloured contours indicate the sea ice edge (as defined by a sea ice concentration of 15 %) for observations (white), CM2-025 (dark blue), CM2-1 (light blue), OM2-025 (dark red), and OM2-1 (light red). The maximum Antarctic sea ice extent is reached in September for CM2-025 and CM2-1 and in August for OM2-025 and OM2-1. The minimum Antarctic sea ice extent is reached in February in all models. The Arctic maximum sea ice extent is reached in March for all models and the minimum in September for all models.

OM2-025 and 110 % for OM2-1. The minimum sea ice extent in summer is with 94 % in CM2-025, 109 % in CM2-1, 93 %
in OM2-025, and 91 % in OM2-1 much better captured by the models in the Arctic compared to the Antarctic. The Arctic is largely surrounded by land which serves as a natural boundary and the integrated sea ice extent can therefore be expected to be more similar to observations. In contrast, the Southern Hemisphere representation of sea ice is much more dependent on the ability of the model to adequately capture processes in the open ocean.

The most noticeable region where the models misrepresent the observed maximum sea ice coverage in the Northern Hemi-
sphere is in the polar North Atlantic where the sea ice in the models spreads too far into the Greenland, Irminger and Labrador Seas (Fig. 10b). In particular the Labrador Sea is covered by sea ice in CM2-1 which is mostly ice-free, i.e. closer to observa-

tions, in CM2-025 and the forced models. The bias in the sea ice coverage is reflected in the MLD which is notably poor in CM2-1 for this region (Fig. 5c). The ACCESS-CM2 models also have too much sea ice in the North Pacific (Bering Sea and Sea of Okhotsk) while the sea ice edge is well captured by the forced ACCESS-OM2 models. The summer decline of sea ice compares well with the observations including the partial opening of Beaufort, East Siberian, and Laptev Seas (Fig. 10d).

Overall, the representation of sea ice in the ACCESS models depends, as expected, to first order whether the ocean-sea ice component is coupled to an atmosphere where sea ice can freely evolve or whether it is forced by an atmospheric reanalysis product that already incorporates sea ice information from observations. The ACCESS-CM2 models underestimate Antarctic sea ice extent all year round but adequately represent the seasonal cycle of sea ice growth and retreat. The rather poor representation of Antarctic sea ice with too little coverage is common amongst CMIP6 models (Beadling et al., 2020). We find that the varying horizontal grid resolution of the ocean-sea ice model component is secondary. The most noticeable difference between the two different ACCESS-CM2 configurations is an even lower maximum Antarctic sea ice extent in CM2-025. A negligible change of simulated Antarctic sea ice in the eddy-permitting regime is also seen in other climate models (Selivanova et al., 2024). Mesoscale ocean dynamics that impact the representation of sea ice are not resolved in the high latitudes at this grid resolution. The ACCESS-CM2 models generally better represent sea ice in the Arctic than in Antarctica. In particular, the summer sea ice extent is well reproduced in the ACCESS-CM2 models compared with Antarctic summer sea ice. The models overestimate the winter sea ice extent with the main biases occurring in the deep convective areas of the polar North Atlantic but also in the Bering Sea and the Sea of Okhotsk, typical for CMIP6 models (Long et al., 2021).

## 3.3 Lower atmosphere: Evaporation

Evaporation from the ocean plays a critical role in global climate. Continental precipitation relies heavily on moisture advected from the oceans, with over 45 % of global land-based precipitation sourced from ocean evaporation (Gimeno et al., 2020). The proportion of land-based precipitation sourced from ocean evaporation is expected to increase with rising global temperatures (Findell et al., 2019). Correct model representation of ocean evaporation is, therefore, important to the simulation of continental precipitation and projecting its likely future change.

Evaporation over the ocean is not directly observed; rather, it is commonly inferred through estimates of air-sea turbulent heat fluxes using data from satellites, buoys and reanalysis datasets. A number of global air-sea turbulent heat flux products exist and have recently been reviewed (e.g., Gutenstein et al., 2021; Tang et al., 2024). Here we evaluate estimates of latent heat flux from CM2-025 and CM2-1 against the IFREMER v4.1 global dataset (Bentamy et al., 2013). In the IFREMER dataset, satellite- and reanalysis-based estimates of surface wind speeds and humidity gradients are used to infer surface turbulent fluxes. The resulting estimates of observed latent heat flux are provided at $0.25°$, monthly resolution from January 1992 to December 2018. We convert latent heat flux ($LHF$) to evaporation ($E$) via $E = LHF/Lv$, where $Lv$, the latent heat of vaporisation, is taken as $2.25 \times 10^6$ J kg$^{-1}$.

Relative to the selected reference (Fig. 11d), on an annual average basis ACCESS-CM2 tends to overestimate ocean evaporation between $60°$ S to $60°$ N, exhibiting a mean anomaly of 0.9 mm day$^{-1}$ (i.e. 28 % increase) in the higher resolution simulation (Fig. 11a) and 0.8 mm day$^{-1}$ (i.e. 26 % increase) in the lower resolution simulation (Fig. 11b). The positive bias

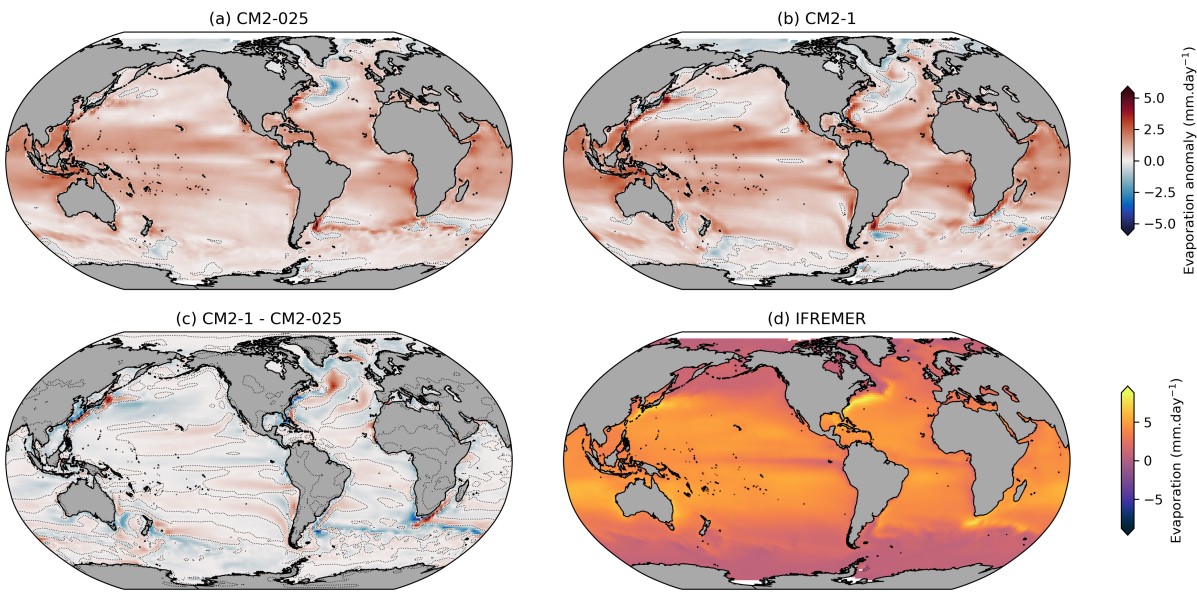

**Figure 11.** Annual average evaporation (mm day$^{-1}$). Anomalies in (a) CM2-025 and (b) CM2-1 relative to (d) IFREMER v4.1 (1992-2018). The difference between the two model simulations is provided in (c). The zero contour is showed as dotted lines in (a) - (c).

dominates in low to mid latitudes (zonal mean bias of more than 1.5 mm day$^{-1}$), with smaller but negative bias dominating in the higher latitudes (zonal mean bias of 0 to -0.5 mm day$^{-1}$), particularly in the Southern Ocean and North Atlantic Ocean (Fig. 11a,b). The difference in estimated evaporation between the higher and lower resolution simulations is modest, with the higher resolution model tending to estimate greater rates of evaporation in the mid latitudes (Fig. 11c). The largest differences in the simulated evaporation pattern emerge in the eddy-active western boundary currents of the North Pacific (Kuroshio), North Atlantic (Gulf Stream) and the Agulhas Current at the southern tip of the African continent. The CM2-025 biases improve in the Kuroshio and Agulhas region when compared with observations. While the evaporation pattern in the Gulf Stream region is also different in the higher resolution model, it does overall not reduce the biases in the region.

## 4    Variability of the coupled system

### 4.1    El Niño-Southern Oscillation (ENSO)

ENSO is a naturally occurring climate fluctuation generated through ocean-atmosphere interactions in the tropical Pacific. ENSO events, which include warming (El Niño events) and cooling (La Niña events) of the central and eastern equatorial Pacific SST anomaly (SSTA), alter the global atmospheric circulation, impacting climate and weather worldwide (McPhaden et al., 2020). These events are often linked to weather extremes like droughts, floods and heat waves in many nations, which also have impacts on agriculture, food security and freshwater resources.

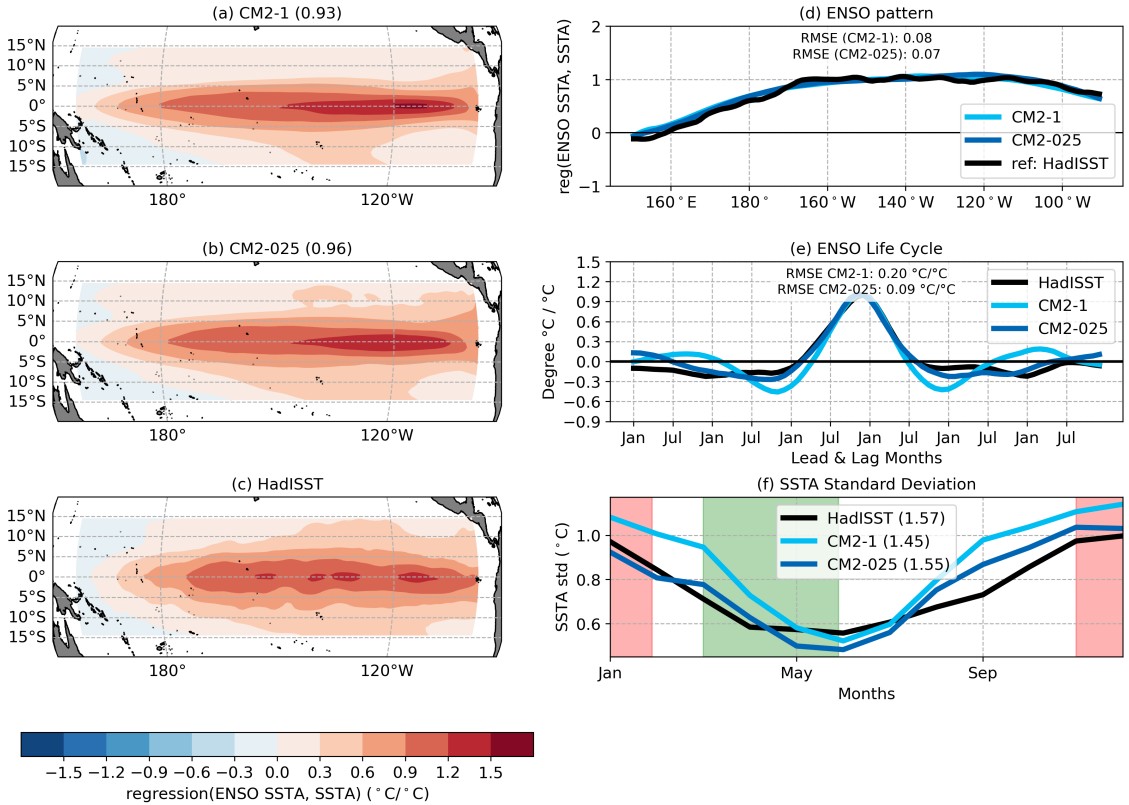

**Figure 12.** Observed and modelled El Niño-Southern Oscillation. (a-c) ENSO pattern, which is defined as the December sea surface temperature anomaly (SSTA) in the Niño3.4 region (120°-170° W, 5° S-5° N) regressed onto December SSTA in the tropical Pacific for (a) CM2-1, (b) CM2-025, and (c) observations (Hadley Centre Sea Ice and Sea Surface Temperature dataset, HadISST). The spatial correlation value between the modelled and observed ENSO pattern is given in the titles of panels (a-b). (d) Meridional average equatorial Pacific (150° E-90° W, 5° S-5° N) SSTA of panels (a-c). (e) ENSO life cycle, where the life cycle is defined by the regression of the December Niño3.4 SSTA index onto the Niño3.4 SSTA over a 36 month lead–lag period. The RMSE displayed in (d) and (e) is the root mean square error between the model and observations. (f) Monthly Niño3.4 SSTA region standard deviation, where the March-May (red) and November-January (green) periods are used to calculate ENSO seasonality are highlighted in colour.

The observed and modelled ENSO pattern is calculated by regressing the December Niño3.4 (region between 120°-170° W and 5° S-5° N) SSTA onto December SSTA in the tropical Pacific (Fig. 12a-c). The observational SST is the Hadley Centre Sea Ice and Sea Surface Temperature dataset (HadISST, Rayner et al., 2003). Visual analysis of Fig. 12a-c reveals that CM2-1 displays the maximum ENSO SSTA too far east. Additionally, the zonal extent of anomalies appears larger in the models as they extend too far to the west. There is no discernible visual change in the modelled ENSO pattern between CM2-1 and CM2-025. This is supported by the related metric, which is defined as the root mean square error (RMSE) of the meridional average (5° S-5° N) equatorial Pacific (150° E-90° W) SST during boreal winter (November-January) between model and observations

(Planton et al., 2020). Both model versions have a similar RMSE value of 0.08° C ° C$^{-1}$ for CM2-1 and 0.07° C ° C$^{-1}$ for CM2-025 (Fig. 12d).

It is widely reported that CM2-1 produces ENSO that mainly displays quasi-biennial oscillations (around 2–2.5 years), instead of the observed more irregular 2–7 year cycles (Rashid et al., 2022; Sullivan et al., 2025). This apparent bienniality is reflected in the life cycle plot by the negative lobes around January the year before and after the event peak (Fig. 12e) and the life cycle metric RMSE of 0.20° C ° C$^{-1}$. The life cycle of CM2-025 displays much weaker negative lobes either side of the event peak, though the lobes remain statistically significant, and a much lower life cycle metric RMSE of 0.09° C °
C$^{-1}$(Fig. 12e). This result suggests that CM2-025 produces ENSO variability with a life cycle much closer to the observations (supported by a Niño3.4 SST power spectrum, not shown), which also suggests a significantly reduced ENSO bienniality.

As in Sullivan et al. (2025), we investigate the biennial tendency with a metric that counts ENSO transitions; i.e., the number of direct El Niño to La Niña in the following year (and vice versa) transitions within 100 years (not shown). Observed direct ENSO transitions were recorded 26 times per century (HadISST data from 1900 to 2014), while CM2-1 records 35 transitions
per century. The higher resolution CM2-025 has a reduced count of these direct ENSO transitions, recording 29 transitions per century. Again, highlighting a reduced biennial tendency of ENSO in CM2-025 compared to CM2-1.

Observed ENSO events also display a clear tendency to be synchronised with the seasonal cycle, which is highlighted by El Niño and La Niña events tending to peak at the end of the calendar year. This seasonal synchronisation is represented in the observed standard deviation of central equatorial Pacific SSTA, which show weak values at the beginning and end of the
ENSO cycle in March-May, and strong values in November-January when the events typically peak (Planton et al., 2020). CM2-1 somewhat reproduces this observed seasonal variance modulation, but there are some clear biases (Fig. 12f). Namely that CM2-1 variance is generally larger than that observed, especially during November-January, and the minimum variance also appears to be shifted later in the calendar year than that observed. CM2-025 appears to generally perform better than CM2-1, in that variance at the ENSO peak is closer to that observed, and the minimum variance is shifted to earlier in the
calendar year which is again more consistent with that observed (Fig. 12f).

## 4.2   Spurious multidecadal variability in the North Atlantic

CM2-025 exhibits spurious multidecadal variability in the North Atlantic that is evident in multiple variables including SST, AMOC transport, sea ice coverage, and MLD (Fig. 13). The fluctuation is strong with SST anomalies in the North Atlantic Ocean of up to 2° C over a 50 year period (Fig. 13a). The anomalies are large enough to influence the global SST (Fig. 1). The
AMOC transport at 26° N varies between approximately 14 and 18 Sv (Fig. 13b). The magnitude of the variability reduces over the length of the simulation but the variability itself does not vanish.

The multidecadal variability in CM2-025 originates from intermittent deep convection in the Labrador Sea with variations also occurring in the deep convective sites in the Nordic Seas north of Iceland. Figure 13c-f shows composite maps for years of anomalous warm and cold SST in the North Atlantic (highlighted in yellow in Fig. 13a). The positive composite represents
years of anomalous warm SST with reduced sea ice coverage and a deeper MLD in the Nordic Seas. Similar to the simulated mean state presented in Sect. 3.1.3, a deep MLD develops in the Labrador Sea. The deep mixed layers indicate enhanced deep

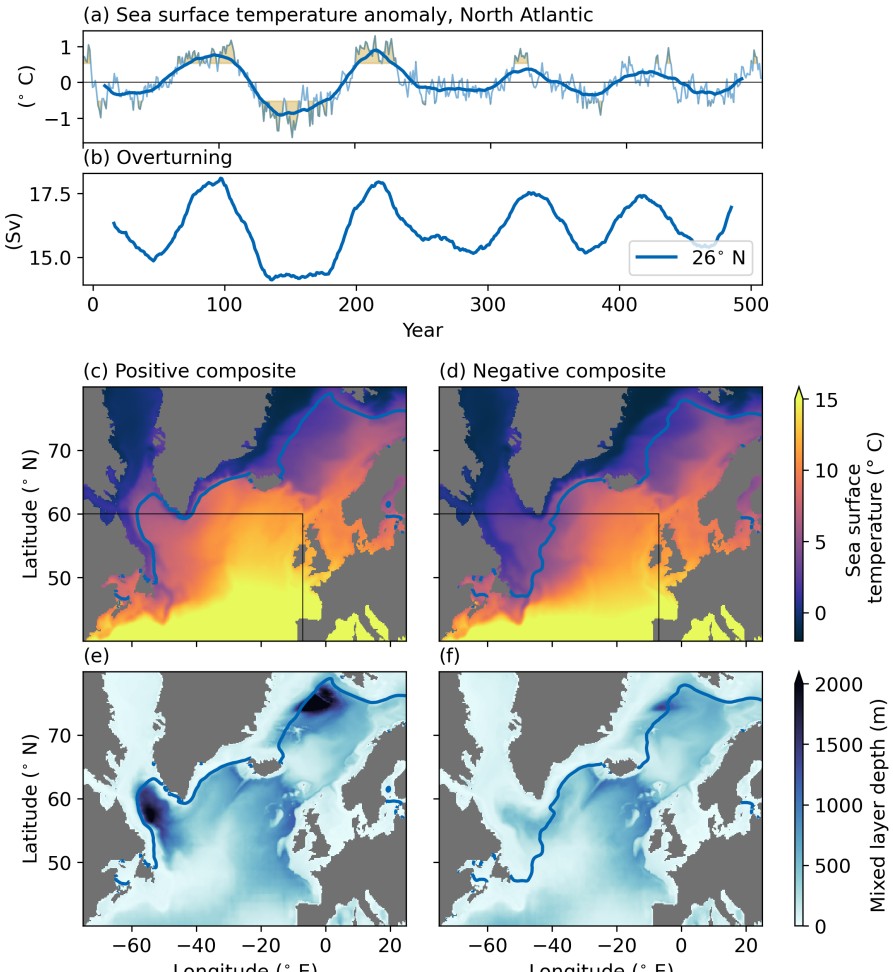

**Figure 13.** Multidecadal variability in the North Atlantic in CM2-025. (a) Time series of sea surface temperature anomalies in the North Atlantic between 25° to 60° N and between 7° to 75° W. The thin line indicates annual averages and the thick line a 30 year rolling mean. Yellow shading represent years with sea surface temperature anomalies outside one standard deviation for which composite maps are plotted in (c-f). (b) Time series of the overturning transport at 26° N as a 30 year rolling mean based on the annual averages. Positive (c and e) and negative (d and f) composites of the multidecadal variability in the North Atlantic. (c-d) Annual mean sea surface temperature and (e-f) March mixed layer depth. The blue contour in (c-f) shows the March sea ice edge. The black box in (c-d) show the northern part of the region (which extends further south to 25° N) over which the sea surface temperature in (a) is averaged.

convection and NADW formation, explaining the larger AMOC transport during this phase (Fig. 13b). The negative composite corresponds to years of cold SST anomalies and reduced AMOC transport. Sea ice extends further east, reducing the MLD in the Nordic Seas and inhibiting deep convection in the fully ice-covered Labrador Sea. The negative composite worsens the
time-mean state in the model with the previously presented biases of too warm SST, missing deep convection in the Labrador

Sea, and too extensive sea ice coverage. The biases are reduced in the positive composite which simulates a climate closer to observations.

The occurrence of a multidecadal variability in the North Atlantic is not unique to ACCESS-CM2, although it does not emerge in the lower resolution version. Meccia et al. (2023) investigate the mechanism for a similar variability that is present in the EC-Earth3 coupled climate model. They find an interplay between freshwater anomalies that accumulate in the Arctic and their release to the North Atlantic, influencing deep convection and subsequently AMOC transport. A strong AMOC carries heat to the North Atlantic, melting sea ice and reducing salinity, which drives freshwater export from the Arctic. The fresher surface layer stabilises the water column in the Labrador and Nordic Seas, ultimately weakening the AMOC. A weakened AMOC reduces heat transport, limits sea-ice melt, and creates salinity anomalies that eventually restore the AMOC cycle. A similar process is described by Jiang et al. (2021) for the IPSL-CM6 model and we assume the same mechanism operates in CM2-025.

There is no apparent reason for why a multidecadal variability in the North Atlantic evolves in some model configurations but not in others. The variability is present in coupled climate models with different submodel components, various grid resolutions, and with different climate forcings (pre-industrial and present day). Meccia et al. (2023) hypothesise a strong dependency on the climate that is simulated by the model. This assumption matches our findings as the lower resolution model CM2-1 does not exhibit a multidecadal variability in the North Atlantic. Mehling et al. (2024) identify a bias in the winter sea ice mean state in the Labrador Sea as a necessary condition to enable strong AMOC variability. The positive sea ice bias in the Labrador Sea in CM2-1 is indicative that AMOC variability is unlikely in this configuration. Similarly, the evolution of an oscillation in the CESM2 model depends on the initial conditions (Danabasoglu et al., 2020), highlighting the sensitivity of the North Atlantic to oceanic conditions. Lastly, the evolution of a multidecadal variability associated with deep convection in CM2-025 is limited to the North Atlantic (Labrador Sea). CM2-025 does exhibit regular deep convection in the Ross and Weddell Seas (Fig. 5, 6) but it does not result in multidecadal variability in the Southern Hemisphere as reported for other models (Held et al., 2019).

## 5 Conclusions

We introduced a new model configuration of the ACCESS-CM2 climate model, CM2-025, with an ocean-sea ice component at a higher grid resolution of 0.25°. The existing lower resolution version, CM2-1, has a 1° ocean grid (Bi et al., 2020). The grid resolution of the atmospheric component of CM2-025 is not changed compared with CM2-1 (N96). With a 0.25° ocean grid resolution in CM2-025, the model is able to explicitly resolve the ocean mesoscale in the low to mid latitudes. We investigate the effect of resolving the ocean mesoscale on the simulated climate by comparing a 500 year present-day simulation of the new CM2-025 model with observations and the lower resolution CM2-1 model. We also compare CM2-025 to uncoupled models from the ACCESS model family at the same two grid resolutions, OM2-025 and OM2-1, and which share the model code for the ocean-sea ice component. The ACCESS-OM2 simulations are forced with a repeat year forcing (May 1990 to April 1991) based on the JRA55-do v1.3 atmospheric reanalysis product (Tsujino et al., 2018). The only differences in the CM2-025 model configuration, besides the higher resolution of the ocean-sea ice component, include, first, a different interpolation

framework (patch in CM2-025 instead of bilinear in CM2-1) for the wind stress to minimise the imprint of the lower resolution atmospheric grid when interpolating onto the finer ocean grid. Second, the bathymetry product used in CM2-025 is improved compared with OM2-025 near the Antarctic continental shelf and at the gateways of the Mediterranean and Baltic Seas. Third, the mesoscale eddy parameterisation scheme is weakened to account for partial resolution of eddies. The lack of any other retuning suggests that, with further efforts, the fidelity of CM2-025 could be further improved.

The higher grid resolution of the ocean-sea ice component in CM2-025 has improved many aspects of the simulated climate. Among these improvements are a faster spin-up of the deep ocean temperatures, a feature that is shared with the forced OM2-025 at the same grid resolution (Fig. 1c). Dynamically, the high eddy-active regions in the mid latitudes, including the strong western boundary currents (Kuroshio, Gulf Stream, and Agulhas Current) and the ACC, are much more energetic (Fig. 4). For this metric, CM2-025 compares best of all four models considered in this study relative to observations. This finding highlights not only the importance of ocean grid resolution but also that the coupled models, that can evolve freely, are more energetic. The impact of the higher grid resolution on the western boundary currents is also reflected in the evaporation pattern that is changed in these regions (Fig. 11). CM2-025 has a much improved ENSO life cycle with a reduced biennality and better representation of the seasonality compared with CM2-1 (Fig. 12e-f). The observed pattern of maximum MLD in the North Atlantic is best captured by CM2-025 (Fig. 5). In particular, CM2-025 is largely sea ice free and able to convect in the Labrador Sea, different to CM2-1 (Fig. 10). CM2-025 also shows reduced deep convection south of Iceland, which is overestimated in CM2-1 and OM2-025. Lastly, due to the improved representation of the bathymetry, the outflow of Mediterranean Sea is better represented (Fig. 8) and biases in SSS of the marginal seas are reduced (Fig. 3).

Various biases present in CM2-1 remain in CM2-025. The errors originate from deficiencies in either the atmospheric model, the oceanic model, or a remaining lack of grid resolution. For example, the mis-representation of clouds and accompanied incoming shortwave radiation over the Southern Ocean in the atmospheric model component lead to too warm SST in the region, which is amplified in CM2-025 over CM2-1 and leads to an even lower summer sea ice extent (Fig. 2, 9 and 10). Too much precipitation over the Bay of Bengal result in too fresh SSS in both coupled model configurations (Fig. 3). Further, erroneous wind patterns over the coastal upwelling regions of the eastern low latitude Pacific and Atlantic also lead to warm SST biases. These biases are much reduced in the uncoupled model. The larger zonal extent of ENSO in CM2-1 remains in CM2-025 (Fig. 12a-c) and comes with a fresh SSS anomaly in the western tropical Pacific (Fig. 3a and c). Key aspects that likely will require even higher ocean grid resolution include biases in the high latitudes – such as open ocean convection in the Southern Ocean and overestimated MLD in the North Atlantic, both of which affect the representation of the MOC (Fig. 5 and 7).

Of particular concern here is the development of a multidecadal variability in CM2-025. This variability is mostly apparent in sea ice coverage and deep convection in the Labrador Sea and also affects the AMOC and global mean surface temperatures. Future studies will have to be cautious when using CM2-025 for the North Atlantic region. While the mean state of CM2-025 is improved over that of CM2-1 (and even over the forced OM2-025) CM2-025 exhibits phases of unrealistic conditions in the Labrador Sea due to this multidecadal variability.

We have presented a model evaluation of a new configuration of the ACCESS-CM2 climate with higher resolution ocean-sea ice component at 0.25°. While the model has an improved representation of the climate in many regions, biases remain. Future work will continue to investigate the role of grid resolution on the simulated climate in the ACCESS-CM2 climate model and to improve each of the submodel components. Planned uses of CM2-025 include a pre-industrial simulation that can serve as a can serve as a spin-up for historical simulations, future scenarios or sensitivity experiments.

*Code and data availability.* The Met Office UM source code and documentation papers are available for use under licence. Information on how to apply for a license are at https://www.metoffice.gov.uk/research/approach/modelling-systems/unified-model (last access: 3 March 2025). The UM and/or JULES code branch(es) used in the publication have not all been submitted for review and inclusion in the UM/JULES trunk or released for general use. CABLE documentation is available at Kowalczyk et al. (2006). The ocean model component MOM5 and the sea ice model component CICE5 are open source. The ACCESS-OM2 configurations are available at https://doi.org/10.5281/zenodo.2653246. Model output to recreate the results presented in this study is available at https://doi.org/10.5281/zenodo.14957592 (Huneke, 2025a). Analysis code is available at https://doi.org/10.5281/zenodo.15061995 (Huneke, 2025b).

*Author contributions.* The technical side of the project was led by MD, with substantial contributions from DB, AH, MO and WH. The model simulation was conducted by WH and MD. Analysis of the output and the writing of the text for the paper was led by WH with substantial contributions from AH, AS, SM, CH and SO.

*Competing interests.* The authors declare that they have no conflict of interest.

*Acknowledgements.* WH is supported by the Australian Research Council Centre of Excellence for Climate Extremes (CE170100023). AH, SM and CH are supported by the Australian Research Council Centre of Excellence for the Weather of the 21st Century (CE230100012). This research was undertaken on the National Computational Infrastructure (NCI) in Canberra, Australia, which is supported by the Australian Commonwealth Government. This research used the ACCESS-NRI's model ACCESS-CM2 infrastructure, which is enabled by the Australian Government's National Collaborative Research Infrastructure Strategy (NCRIS). We thank the COSIMA consortium (http://cosima.org.au) for technical support and the development of the ACCESS-OM2 model configurations used. Part of ACCESS modelling work was jointly funded through CSIRO and the Earth Systems and Climate Change Hub and subsequent Climate Systems Hub of the Australian Government's National Environmental Science Program (NESP2). ACCESS simulations, data processing and data publication were undertaken with the assistance of resources from NCI Australia, a NCRIS-enabled (National Collaborative Research Infrastructure Strategy) capability supported by the Australian Government.

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
