# Peer review of "The ACCESS-CM2 climate model with a higher resolution ocean-sea ice component $(1/4^{\circ})$"

_EGUsphere, 2025_

## Author Comment (AC3)

**Response to reviewer comments**

We sincerely thank the editor, Riccardo Farneti, and the three reviewers (Brandon Reichl, Mitchell Bushuk, and one anonymous reviewer) for their efforts to thoroughly review our manuscript. The reviews were generally positive and consistent; the primary concern was the need for more detail on the representation of the mesoscale across model configurations. We detail here responses to the comments and indicate changes made to the manuscript. The reviewers' comments have been listed in order below in italic text, followed by our responses in normal text, and new manuscript text in blue.

New Figures and Figures that support our responses have been added to the end of this response document (labelled Fig. R1, R2, etc).

**Reviewer 1**

Reviewer 1: This manuscript introduces and evaluates a new configuration of the Australian Community Climate and Earth System Simulator (ACCESS-CM2) coupled model, based on a 1/4-degree ocean-sea ice component (ACCESS-CM2-025).

The primary objective of developing this new model was to enhance the representation of ocean mesoscale features and broaden the scope of climate modeling research applications. The manuscript comprehensively assesses the simulated climate of ACCESS-CM2-025 over a 500-year present-day simulation by comparing it against various observation-based datasets, the existing lower-resolution version of the same coupled model (ACCESS-CM2-1), and two forced ocean-sea ice models (ACCESS-OM2), OM2-025 (0.25° resolution) and OM2-1 (1° resolution), which share the same ocean-sea ice components as the coupled models but are driven by JRA-55-do.

Overall, the study concludes that the higher grid resolution of the ocean-sea ice component in CM2-025 has improved many aspects of the simulated climate, especially in dynamically active regions. Still, some biases persist, pointing to remaining deficiencies in the atmospheric or oceanic model components, or a need for even higher resolution. The manuscript is well-written, and the authors effectively present the evaluation of this new coupled system using key metrics. My main suggestion, detailed below, is to include additional information about the ‡-degree ocean setup and to clarify the relative contributions of increased horizontal resolution versus mesoscale parameterizations to some of the observed improvements. The manuscript is well-suited to Geoscientific Model Development, and I strongly recommend publication pending minor clarifications as outlined below.

**Reply**: Thank you for these encouraging remarks in support of this manuscript.

**Specific comments**

**Reviewer 1:** Lines 46-48: Please briefly list what are the different approaches that the modeling centers take to the issues of eddy-permiting resolutions. Which of these approaches does ACCESS-CM2-025 follow?

**Reply**: ACCESS-CM2-025 and ACCESS-OM2-025 both follow the same approach to applying eddy parameterisations at this eddy-permitting resolution. This approach involves applying both GM and Redi diffusivities, but with a smaller value for the coefficient of diffusivity than for the 1° configuration. We have added this information to section 2.1 (see responses below) since the ACCESS models have not been introduced in the manuscript the line this comment is referring to.

**Reviewer 1:** Table 1: It would be helpful to include the total number of processors used for each configuration, along with the breakdown across model components. Additionally, the manuscript should clarify why performance differs so substantially between OM2-1 and OM2-025 compared to CM2-1 and CM2-025.

**Reply**: We have added the cores (total + for each model component) to Table 2. The core distribution in CM2-1 results in long wait times for coupler communication, where

the runtime for the atmosphere is much longer compared with the ocean. The different core distribution in CM2-025 reduced such waiting times, resulting in a similar performance compared with CM2-1. We have added this information to the table caption:

The core count in CM2-025 allows for similar runtimes across model components and minimal waiting for coupler communication, resulting in similar performance compared with CM2-1, which is slowed down by the atmosphere.

Reviewer 1: Lines 124-125: Please briefly describe the sub-grid scale parameterizations used in this study, including both mesoscale and submesoscale schemes, as well as the vertical mixing scheme employed. In particular, given that modeling centers differ in whether Gent-McWilliams/Redi is applied at this resolution, it would be helpful to clarify the approach taken here. Additionally, please explain how liquid and frozen freshwater inputs were represented or distributed in the model.

**Reply**: As noted above, we use a weak GM/Redi parameterisation at these eddy-permitting resolutions. Otherwise, parameterisations (for vertical mixing, submesoscale and freshwater inputs) are independent of resolution. We have added a new paragraph to the end of section 2.1 to include these details:

The primary sub-grid-scale parameterisations in ACCESS-CM2 and ACCESS-OM2 are similar at each resolution, including the submesoscale parameterisation (Fox-Kemper et al., 2008), biharmonic isotropic Smagorinsky horizontal friction (Griffies and Hallberg, 2000) and KPP vertical diffusivity (Large at al., 1994). The background diffusivity is set to zero at 0.25°, due to excessive numerical mixing, while the Jochum (2009) latitudinally-dependent, depth-independent background vertical tracer diffusivity scheme is used at 1°. (Details of the implementation of these parameterisations are provided by Kiss et al. 2020). The parameterisation of mesoscale eddies is applied more judiciously, to account for the partial resolution of mesoscale features at 0.25°. The Redi diffusivity for lateral isopycnal mixing is set to a constant 600 m2s-1 at 1° resolution, but is scaled by the grid resolution relative to either the first baroclinic Rossby radius, or the equatorial Rossby radius for latitudes between ±5°N, with a maximum diffusivity of 200 m2s1. The GM diffusivity is depth-independent but varies laterally, scaled by a combination of an inverse Eady growth rate timescale and a length scale squared (50 km at 1° and 20 km at 0.25°), as well as the same grid scaling used in the calculation of Redi diffusivity. There is a maximum diffusivity of  $600 \text{ m}^2\text{s}^{-1}$ at  $1^{\circ}$  resolution and  $200 \text{ m}^2\text{s}^{-1}$  at  $0.25^{\circ}$  resolution. The reduced eddy parameterisation at 0.25° has been tuned in ocean-sea ice simulations with a biogeochemical model to optimise transport of tracers by the ocean's overturning circulation.

Liquid and frozen freshwater input is passed from the land model via the coupler to the sea ice model and then to the ocean. Accumulated land ice over Greenland and Antarctica is discharged as ice berg calving, liquid freshwater is discharged as river runoff. We have not added any new text to the manuscript since details on liquid and frozen freshwater are given in Bi et al. (2020) and Kiss et al. (2020) and no relevant changes have been implemented in CM2-025.

**Reviewer 1:** Lines 138-140: Based on the slope of the ocean temperature time series, it appears that the top-of-atmosphere energy imbalance differs substantially between CM2-1 and CM2-025, with CM2-025 showing a significantly reduced imbalance. Since the only change

between these configurations is the ocean and sea ice horizontal resolution (with no additional atmospheric tuning), this is a notable result that the authors may wish to emphasize more clearly.

**Reply**: We have added this result to the text as follows:

The surface and interior ocean warming (through ocean heat uptake) is reflected in energy imbalances at the top of the atmosphere (Bi et al., 2020) which is notably reduced in CM2-025 compared with CM2-1 (compare slopes in Fig. 1c).

Reviewer 1: Lines 215–217: The statement, "The fact that both higher-resolution models exhibit the same behaviour confirm the requirement of a higher grid resolution to adequately resolve the spatial structure of the ACC, even if the integrated transport deviates more from the observational estimate," may overstate the attribution to resolution alone. Given that the GM parameterization is also applied, how can the authors be certain that the improved ACC structure is solely due to increased resolution? Clarifying the role of GM in this context would strengthen the argument.

**Reply**: As noted above (and included in the revised manuscript), the GM diffusivity is weaker at 0.25°, and is scaled by grid resolution. Therefore, the 0.25° models have a stronger eddy field in the ACC, better representing the interfacial form stresses that dictate the vertical transfer of momentum within the ACC (Ward & Hogg, 2011). We have modified this sentence to read:

The fact that both higher resolution models exhibit the same behaviour confirms the requirement of a higher grid resolution and lower eddy diffusivity to better represent the eddying structure of the ACC, and associated vertical momentum transfer (Ward and Hogg, 2011), even if . . .

Reviewer 1: Table 2 and Figures 2 and 3: Is the thickness of the first model layer, used for extracting SST and SSS, the same as that of the corresponding level in the WOA13 vertical grid? If not, were any interpolations applied to facilitate the comparison? Additionally, please include a reference in Table 2 to the observation-based MLD dataset used.

**Reply**: The WOA13 observations were interpolated onto the model grid, the SST and SSS then represent the uppermost value.

We have added the reference to the MLD dataset in Table 2.

Reviewer 1: Lines 256–263: This paragraph mixes observational and model definitions of MLD, making it difficult to identify the key point. The observational dataset referenced (https://www.seanoe.org/data/00806/91774/) uses an updated MLD definition based on a 0.03 kg/m³ density difference from 10 m depth, as described in Treguier et al. (2023). Therefore, the appropriate citation should be de Boyer Montégut et al. (2023) rather than the original 2004 paper. In contrast, the model uses a buoyancy threshold of 0.0003 m/s² relative to the surface, following Griffies et al. (2016). It is unclear how these two criteria are "nearly identical," as claimed. A clearer justification or quantification of their similarity would be helpful.

**Reply**: We have corrected the reference for the observational dataset to de Boyer Montégut et al. 2023 (here and in Table 2).

Assuming buoyancy is given as  $\Delta b = \frac{g\Delta\rho}{\rho_0}$  with  $g = 9.8 \text{ m s}^{-2}$ ,  $\rho_0 = 1035 \text{ kg m}^{-3}$  and  $\Delta b = 0.0003 \text{ m s}^{-2}$ , the buoyancy-based definition gives  $\Delta \rho = 0.0317 \text{ kg m}^{-3}$ . This is within 5% of the density definition used in the observational product. We have updated the text to report this number:

The buoyancy-based definition used in the models approximates the density-based definition used in the observational product to within 5%. For the purpose of this model validation, where we focus on the bias of the maximum MLD, the errors introduced due to the different choice of MLD definition is secondary as the models present large deep biases and observations tend to overestimate rather than underestimate MLD (de Boyer Montégut et al., 2004).

Reviewer 1: Lines 263–269 and Figure 5: The authors compare the maximum monthly climatological mixed layer depth rather than analyzing a specific month or season. While this approach highlights the deepest annual MLD, it may obscure important aspects of the seasonal cycle, such as the timing and rate of MLD deepening and shoaling. Including an assessment or brief discussion of how well the seasonal evolution of MLD is represented in the models, particularly in regions with strong seasonal variability, would strengthen the analysis. Given the presence of open-ocean polynyas in the models, a focus on a specific month or season would also be especially informative. Finally, a bias map (model minus observations) would add value, as Table 2 only covers a limited number of regions.

Reply: We decided to continue displaying the maximum bias of the MLD in Figure 5 (which now shows a bias map) as this maximum bias translates to biases in the interior ocean, independently when in the year the bias might occur. We have added a new Figure 6 (Fig. R1 in this response document) showing the seasonal evolution of the maximum monthly MLD for the different regions reported in Table 2. The findings from this new figure have been added throughout Section 3.1.3, we list below the most relevant sentences:

Figure 6 shows the seasonal evolution of the maximum monthly climatological MLD for selected regions.

Both the coupled and uncoupled models overestimate the maximum MLD in regions of deep MLD while summertime values are comparable (Fig. 5a-c, Fig. 6a-c, g-h).

Both OM2 models exhibit a deeper maximum wintertime MLD between June and November. The deepening occurs within about a month and then remains relatively constant through winter until November. In contrast, the CM2 models display a more gradual (and realistic) deepening, reaching their maximum in October. The faster MLD deepening in the OM2 models, which is not limited to the MLD evolution in the Weddell and Ross Seas (Fig. 6), likely results from the infinite heat capacity of the prescribed atmosphere, which enables rapid ocean heat loss. The MLD evolution is slower in the CM2 models as the atmosphere must adjust to the heat flux from the ocean.

The CM2 models exhibit slight improvements both for the peak wintertime MLD value and for the seasonal evolution of MLD.

Reviewer 1: Lines 270–277; lines 295-296: It is surprising that open-ocean polynyas occur in the 1/4° model. Were these events persistent, and how long did they typically last? It would be helpful to clarify whether the authors attempted to enhance restratification processes, either

through mesoscale or submesoscale parameterizations, during model development to reduce or eliminate the occurrence of such features.

Reply: Excessive open ocean convection in the Southern Ocean at 0.25° is, unfortunately, not unique to ACCESS and also emerges in other models with a higher ocean grid component (Heuzé et al., 2021). We have added this information to the text as follows: Open ocean polynyas have been reported twice in the Weddell Sea for the observational period (Gordon, 1978; Carsey, 1980; Campbell at al. 2019), but are a common feature in most climate models at various grid resolutions, including ACCESS-CM2 (Heuze et al., 2021).

Deep convection in the Ross and Weddell Seas occurs regularly in the ACCESS models, even at 0.25°. Figure R2 shows the annual maximum MLD for CM2-1 and CM2-025 for both regions. At the end of section 4.2, we now mention that deep convection in the Weddell and Ross Seas occurs regularly and is not subject to multidecadal variability:

Lastly, the evolution of a multidecadal variability associated with deep convection in CM2-025 is limited to the North Atlantic (Labrador Sea). CM2-025 does exhibit regular deep convection in the Ross and Weddell Seas (Fig. 5, 6) but it does not result in multidecadal variability in the Southern Hemisphere as reported for other models (Held et al., 2019).

Please see the response to the last comment regarding the point on persistence. The new Figure 6 showing the seasonal evolution of the MLD in different regions, including the Weddell and Ross Seas, and the accompanying new text address this comment.

We agree that a comment on retuning is warranted. The tuning of the ocean part of ACCESS-CM2-025 has been largely achieved by previous tuning of ACCESS-OM2-025. Additional tuning of topography in key straits (which emerges in coupled modes but is controlled by salinity restoring in ocean-sea ice configurations) was done, but these minor modifications tend not to alter the large-scale circulation. We have added the following statement to the end of the first paragraph of the conclusion to highlight the changes in eddy parameterisation and to clarify the potential for further retuning:

Third, the mesoscale eddy parameterisation scheme is weakened to account for partial resolution of eddies. The lack of any other retuning suggests that, with further efforts, the fidelity of CM2-025 could be further improved.

**Reviewer 1:** Figure 6: The thick tick marks on the x-axis are misaligned between the upper and lower panels. Which one is correct? Based on the latitude labels in the lower plots, the transitions between ocean basins do not appear to occur at 30°S as indicated.

**Reply**: The marks on the upper plots (0.25 models) were indicating different ticks. We have corrected the ticks on the x-axis. The transition between the full Southern Ocean and the Atlantic Basin occurs at 34°S, we have corrected this in the figure caption and manuscript text. We have also added a vertical thick line and text annotations to further highlight the two regions.

**Reviewer 1:** Line 296: In addition to highlighting improvements to the vertical mixing scheme, the authors might also consider discussing improvements to re-stratification processes via mesoscale and submesoscale parameterizations.

**Reply**: We have changed the text to:

Improving the representation of MLD might therefore require a much higher grid resolution

in both the horizontal and vertical (Kiss et a., 2020) as well as improvements to the vertical mixing schemes and mesoscale and submesoscale parametrisations which help to set the interior stratification by re-stratifying the water column in winter.

**Reviewer 1:** Lines 319–321: The authors might consider including a comparison of the AMOC vertical structure in depth space at 26°N, where long-term observational data are available (e.g., from the RAPID array), to provide a more direct evaluation against observations.

**Reply**: We have included a discussion of where in the water column the maximum transport occurs to the second paragraph of Section 3.1.4 as follows:

The strength of the upper cell at 26°N can be compared to available observations (McCarthy et al., 2015) by calculating the maximum transport in the vertical (where density is the vertical coordinate for the models and depth is the vertical coordinate for the observations). The observational estimate of 17.2 Sv is underestimated in all ACCESS configurations (Table 2)...

Depth estimates of the simulated maximum transports are obtained by converting the model overturning from density to depth coordinates. The observed peak transport occurs at a depth of 1030 m and is biased shallow in the coupled models (782 m in CM2-025, 969 m in CM2-1), matches the observed depth well in OM2-025 (1030 m) and is biased deep in OM2-1 (1173 m).

**Reviewer 1:** Lines 429–431 and Figure 10: I do not clearly see the "smaller but negative bias dominating in the higher latitudes" as stated. Please double-check this claim for accuracy. Additionally, consider highlighting the zero-contour line in Figure 10 to make the sign and structure of the bias more apparent.

**Reply**: Thank you for the suggestions to improve Figure 11 (previously Figure 10) and its associated text. Figure 11a-c has been updated to include a zero-contour line. To clarify the statement around bias in the higher latitudes, we include the zonal mean bias in Figure R3. From the figure we see that biases tend toward negative values in the higher latitudes, albeit with smaller magnitudes ( $\sim 0$  - -0.5 mm.day-1) compared to positive biases in the lower latitudes ( $\sim 0$  - 1.5 mm.day-1). We have amended the text to include the zonal mean bias values:

The positive bias dominates in low to mid latitudes (zonal mean bias of more than 1.5 mm day-1), with smaller but negative bias dominating in the higher latitudes (zonal mean bias of 0 to -0.5 mm day-1), particularly in the Southern Ocean and North Atlantic Ocean (Fig. 11a,b).

**Editorial/Typographical Comments**

Reviewer 1: Line 216: "confirm" = confirms.

Reply: Changed.

**Reviewer 1:** Figure 8: Please consider increasing the size of the two plots to improve readability and better highlight the differences between the model configurations.

**Reply**: We have changed the aspect ratio to improve readability of the figure.

**Reviewer 1:** Lines 338-339: "(fresh bias in the Southern Hemisphere for CM2-025)", this is misleading as CM2-1 shows a near-surface fresh bias in the Southern Hemisphere. Please clarify.

**Reply**: The text here discussed the biases in the subsurface ocean and we therefore do not mention the fresh surface bias in CM2-1. We have added the word subsurface to the text when mentioning the CM2-025 fresh bias for clarity:

The warm bias goes along with high salinity values (fresh subsurface bias in the Southern Hemisphere for CM2-025) and exists in all models, but is more pronounced in the coupled models with some improvements in CM2-025 over CM2-1.

**Reviewer 1:** Lines 368-369: Please improve this sentence; "... CM2-1 rather poor" perhaps should be "... CM2-1, which is rather poor".

**Reply**: Changed to: The representation of the minimum summer sea ice extent is 11 % for CM2-025 and 33 % for CM2-1, which is rather poor.

Reviewer 1: Line 434: "... and Aghulas.." = "The CM2-025 biases improve in the Kuroshio and Agulhas region when compared with observations." (Spelling of "Agulhas").

**Reply**: We fixed all occurrences in the manuscript, thank you for spotting this spelling mistake.

**Reviewer 1:** Line 519: "...onto the finder ocean grid." = "...onto the finer ocean grid."

Reply: Fixed

**References:**

Treguier, Anne Marie, Clement de Boyer Montégut, Alexandra Bozec, Eric P. Chassignet, Baylor Fox-Kemper, Andy McC. Hogg, Doroteaciro Iovino et al. "The mixed-layer depth in the Ocean Model Intercomparison Project (OMIP): impact of resolving mesoscale eddies." Geoscientific Model Development 16, no. 13 (2023): 3849-3872.

**Reviewer 2**

Reviewer 2: This manuscript provides a model documentation of a new ACCESS CM2-025 coupled climate model, which has an increased ice-ocean resolution of \$\frac{1}{4}\$ degree and uses the same atmosphere land components as ACCESS CM2-1 (which has a 1 degree ice-ocean resolution). The manuscript compares \$4\$ simulations each run for 500 years: Year-2000 control experiments performed with the two coupled models and ice-ocean simulations forced with repeat year forcing performed with the ice-ocean components of each coupled model. The authors compare these four simulations to assess the impact of increasing the ocean and sea ice resolution from 1 to \$\frac{1}{4}\$ degree. They find some simulation benefits associated with moving to \$\frac{1}{4}\$ degree resolution, including increased eddy kinetic energy, improved ENSO dynamics, and reduced model drift. However, a number of climate biases are unchanged, and others are degraded. They also find that the CM2-025 model exhibits a large centennial-scale variability associated with deep convection in the North Atlantic.

The paper is clearly written, easy to follow, well referenced, and has a number of results that will be of interest to the broader modelling community. The subject matter is certainly appropriate for GMD. I appreciated that the paper was well scoped, which made it very readable for a model documentation paper. I have a number of minor comments for the authors to consider, and also one major comment related to the sea ice analysis in the manuscript. I am recommending major revisions, since I consider the sea ice comment essential to address, but I expect that most of my other comments should be relatively straightforward to address. Also, note that I only read the comments of Reviewer 1 after doing this review, in an effort to provide an independent assessment. I agree with their assessment, and there are some commonalities between our reviews.

**Reply**: Thank you for the supportive review and providing further suggestions to improve the manuscript.

**Major comment**

**Reviewer 2: Sea ice analysis:**

Section 3.2: The observed sea ice extent values look strange, both for the Arctic and Antarctic. Which NSIDC observational product is being used here? Was the polar hole in the Arctic taken into account? If I compute sea ice extent climatologies using the NSIDC SIC CDR dataset (https://nsidc.org/data/g02202/versions/5), I get the following approximate values, which are notably different from those shown in Fig. 8 and Table 2:

Arctic (Max, Min): 15, 6.5

Antarctic (Max, Min): 19, 4.5

Please double check the sea ice observational values, as these large differences have important implications for the model results. This will affect many of the statements throughout section 3.2, for example Lines 367-369, 372-375, and 385-390.

**Reply**: Thank you for spotting the inconsistency with the reported sea ice extend. We accidentally reported on the sea ice area (for observations & models) and mislabelled it

as sea ice extent. We have corrected this and are now presenting the sea ice extent and have updated Table 2 and Figure 9 (previously Figure 8) accordingly. There are still some inconsistencies between the values provided for the observations in the reviewer comment and the values we present; we contribute the differences to the fact that we interpolate the observations to the CM2-025 grid. Because we accidentally reported the sea ice area instead of sea ice extent for the observations as well as for the models, the statements in section 3.2 largely still hold. We have made the following corrections to the text:

The maximum sea ice extent in winter reaches 84 % for CM2-025 and 93 % for CM2-1 of the observed extent (Table 2). The representation of the minimum summer sea ice extent is 11 % for CM2-025 and 33 % for CM2-1, which is rather poor.

The ACCESS-OM2 models capture the magnitude of the sea ice maximum well (97 % for OM2-025 and 95 % for OM2-1) although the sea ice grows faster with the maximum occurring a month earlier than in the observations.

The coupled model configurations simulate 120 % for CM2-025 and 125 % for CM2-1 of the observed maximum sea ice extent. The forced models have a very similar maximum sea ice extent to CM2-025 of 108 % for OM2-025 and 110 % for OM2-1. The minimum sea ice extent in summer is with 94 % in CM2-025, 109 % in CM2-1, 93 % in OM2-025, and 91 % in OM2-1 much better captured by the models in the Arctic compared to the Antarctic.

**Minor comments**

Reviewer 2: L37-38: This is a bit confusing since the resolution is the inverse of grid spacing, thus has different units than the Rossby radius. Suggest changing to something like: "A lateral grid spacing of less than half of the Rossby deformation radius is needed..."

**Reply**: Changed accordingly.

**Reviewer 2:** L71: "...from the atmosphere to the ocean on larger scales..."

Reply: Changed accordingly.

**Reviewer 2:** L70-81: Also suggest mentioning that sufficiently high atmospheric resolution is required to capture these small-scale coupled air-sea interactions.

**Reply**: We have included this information to the paragraph as follows:

Oceanic fronts, and the associated temperature gradients, are more sharply defined in eddying models which is why at least eddy-present models, together with a sufficiently high atmospheric resolution, are required to capture these mechanisms (Tsartsali et al., 2022).

**Reviewer 2:** Table 1: It is useful to know the performance in simulated years per day of each configuration, but it would also be helpful to know the number of CPU cores required for these simulations. Please add another column providing this.

**Reply:** We have added the cores (total + for each model component) to Table 2.

**Reviewer 2:** L115: I assume that the sea ice model uses an ice-thickness distribution? Please add the number of ice thickness categories here.

**Reply**: We have added the number of ice thickness categories to the text: Both configurations have 50 layers in the ocean using a z\* vertical coordinate system and five ice thickness categories (plus open water).

**Reviewer 2:** L121-124: Since the impacts of these bathymetry changes come up a number of times in the manuscript, it would be helpful to provide some additional details on the exact changes that were made (i.e. the number of grid cells changed, the change in seafloor height, etc).

Reply: We have revised the description of the bathymetry products used. The previous text was perhaps misleading as CM2-025 uses a different bathymetry product which has been corrected for narrow straits using the same values as in OM-025. Additionally, a few more fixes had to be made, which included e.g. deepening of approximately 150 grid cells at the opening of the Baltic Sea to 35 m. However, we do believe that reporting on every single change is not a meaningful analysis for this paper and we therefore decided not to include more details on the depth and number of grid cells changed. We have corrected the description of the bathymetry products to:

The bathymetry for CM2-1 uses a legacy dataset (Bi et al. 2013) while OM2-1 and OM2-025 use a modified version of the GFDL-CM2.5 bathymetry (Griffies et al. 2015, Kiss et al. 2020). CM2-025 uses a new topography created from GEBCO2014 v20150318 30 arc-second topography (GEBCO, 2014) which is revised in specific locations (key straits) by copying depth values from the OM2-025 bathymetry. Additional key locations also required revision including a change of the land mask around the Antarctic continental shelf to avoid model crashes during local storm events. Storms can occasionally grow stronger in the coupled system compared with the prescribed atmosphere in the OM2 models (see Section 2.2), hence it is not an issue in OM2-025. Other revisions involve the deepening of the gateways of the Mediterranean and Baltic Seas which otherwise salinify or freshen in the coupled model due to limited exchange between the respective marginal sea and the open ocean. The grid cell depth was chosen primarily to improve representation of water mass exchange, rather than to enhance overall realism.

Reviewer 2: L125-126: Given the strong focus on eddy impacts in this manuscript, it would be helpful to provide more details here on the GM implementation. For example, what diffusivity coefficients are used in the various simulations? Is a lateral resolution function employed? Is there a vertical structure function used? Where is the GM parameterization active in the 0.25 degree runs?

Reply: This information was also requested by Reviewer 1 and 3, and we have added a new paragraph in Section 2.1 to include these details. In short, we use a weak GM/Redi parameterisation at the eddy-permitting resolutions, and explain the details in this new text: The primary sub-grid-scale parameterisations in ACCESS-CM2 and ACCESS-OM2 are similar at each resolution, including the submesoscale parameterisation (Fox-Kemper et al., 2008), biharmonic isotropic Smagorinsky horizontal friction (Griffies and Hallberg, 2000) and KPP vertical diffusivity (Large at al., 1994). The background diffusivity is set to zero at 0.25°, due to excessive numerical mixing, while the Jochum (2009) latitudinally-dependent, depth-independent background vertical tracer diffusivity scheme is used at 1°. (Details of

the implementation of these parameterisations are provided by Kiss et al. 2020). The parameterisation of mesoscale eddies is applied more judiciously, to account for the partial resolution of mesoscale features at  $0.25^{\circ}$ . The Redi diffusivity for lateral isopycnal mixing is set to a constant  $600 \text{ m}^2\text{s}^{-1}$  at 1° resolution, but is scaled by the grid resolution relative to either the first baroclinic Rossby radius, or the equatorial Rossby radius for latitudes between  $\pm 5^{\circ}\text{N}$ , with a maximum diffusivity of  $200 \text{ m}^2\text{s}^1$ . The GM diffusivity is depth-independent but varies laterally, scaled by a combination of an inverse Eady growth rate timescale and a length scale squared (50 km at 1° and 20 km at  $0.25^{\circ}$ ), as well as the same grid scaling used in the calculation of Redi diffusivity. There is a maximum diffusivity of  $600 \text{ m}^2\text{s}^{-1}$  at  $0.25^{\circ}$  resolution. The reduced eddy parameterisation at  $0.25^{\circ}$  has been tuned in ocean-sea ice simulations with a biogeochemical model to optimise transport of tracers by the ocean's overturning circulation.

**Reviewer 2:** Section 2.1: Do the ice-ocean simulations include a surface salinity restoring? If so, please add the restoring data set used and the restoring strength.

**Reply**: Yes, there is surface salinity restoring in the ACCESS-OM2 models. We added additional information on the surface salinity restoring at the end of the third paragraph in Section 2.1:

The sea surface salinity in ACCESS-OM2 is globally relaxed toward the World Ocean Atlas 2013 v2 (WOA13, Zweng et al. 2013, same as used for initial conditions, see Section 2.2) through a salt flux, with the strength determined by a piston velocity of 33 m per 300 d (Kiss et al., 2020). The restoring salt flux is derived from the difference between surface salinity in the model and in WOA13 and kept below  $\pm 0.5$  psu to prevent excessively large fluxes.

Reviewer 2: L128-136: L128 states that the models are initialized from a prior PI control simulation, then L132 says that WOA is used. Which one is it? This should be clarified.

**Reply**: CM2-1 is initialised from the existing CM2-1 pre-industrial simulation from Bi et al. (2020). The atmospheric component of CM2-025 is also initialised from this pre-industrial simulation, however, the ocean is initialised from WOA13 observations. We have clarified this in the text, which now reads as:

All simulations are integrated over 500 years. Both CM2-1 and CM2-025 use present-day forcing representing an average of the 1985-2014 (nominally year 2000) atmospheric conditions. Using present-day forcing allows for comparison of the model output to observations which are lacking the required spatial coverage for the pre-industrial climate which is often applied for climate model control simulations. CM2-1 is initialised from the 950 year-long pre-industrial CM2-1 simulation documented in Bi et al. (2020). CM2-025's atmospheric component is also initialised from the 950 year-long pre-industrial CM2-1 simulation while the ocean component in CM2-025 is initialised from observations. The initial conditions for the ocean-sea ice component are the same as for the forced OM2 configurations: The OM2 (and CM2-025) configurations are initialised from...

**Reviewer 2:** L139: The ocean heat uptake and interior ocean warming also results from TOA energy imbalance. This should also be mentioned here.

**Reply**: We have reworded the text to:

The surface and interior ocean warming (through ocean heat uptake) is reflected in energy imbalances at the top of the atmosphere (Bi et al., 2020).

Reviewer 2: L137-148: Given that this is a year 2000 control, we expect SST and interior ocean warming due to the TOA energy imbalance. It would be helpful to have a sense of how much of the model drifts shown in Figure 1 are due to the Year-2000 forcing and how much are due to erroneous drift associated with model physics and numerics errors. Were any 1850 pre-industrial control runs performed with these configurations? Comparing these to the Year-2000 control would be a way to estimate the contributions from Year-2000 forcing in Figure 1. If any pre-industrial control runs exist, I suggest looking at them and adding some comments on this in the text.

**Reply**: We have compared the drift in the SST and interior ocean temperature in CM2-025 under present-day forcing with a shorter (90 year) test simulation of CM2-025 under pre-industrial conditions (Figure R4). The interior ocean temperature trend is reduced to 30% in the pre-industrial run, implying a large role of the applied forcing for the resulting temperature drift. We have added this information to the text as follows:

The interior ocean temperature trend in a 90-year test simulation of CM2-025 under preindustrial conditions is reduced to 30% compared with the present-day CM2-025 simulation presented here, suggesting a substantial contribution of the present-day forcing to the total drift.

Reviewer 2: Table 2: I don't see a superscript "b" in this table.

**Reply**: We have corrected the superscripts in Table 2.

Reviewer 2: L164-168: It seems that the ocean component also plays a role in the Southern Ocean SST biases. In particular, the ocean-only OM2-025 run displays a number of Southern Ocean SST bias features that are also present in the CM2-025 run, albeit with smaller magnitude, suggesting that ocean physics errors are also a contributor to these biases. I suggest mentioning this point here and softening the current text, which makes a too-definitive attribution to the atmospheric component.

**Reply**: We have softened the text and added information that ocean physics might also impact the Southern Ocean SST biases:

In particular, the warm bias in the Southern Ocean is predominantly due to too little cloud coverage and therefore too much incoming shortwave radiation, a common issue amongst climate models (Hyder et al., 2018). The ocean-only OM2-025 configuration displays a number of Southern Ocean SST bias features that are also present in the CM2-025 run, albeit with smaller magnitude, suggesting that ocean physics errors are also a contributor to these biases (see end of paragraph).

Reviewer 2: L187-190: Suggest citing Khosravi et al. (2022) here (https://agupubs.onlinelibrary.wiley.com/doi/10.1029/2021EF002282), who show that similar positive salinity biases in the Canadian Basin are ubiquitous across CMIP6 models.

**Reply**: We added a sentence: Similar positive salinity biases in the Canadian Basin are ubiquitous across CMIP6 models (Khosravi et al., 2022).

**Reviewer 2:** Line 190: Change to: "Other challenging regions..."

Reply: Changed.

Reviewer 2: Table 2: I suggest adding RMSE values for SST and SSS to Table 2 (i.e. the RMSE based on local SST/SSS errors). This would help with, for example, the point made on L195-196.

**Reply**: We have added the RMSE for both SST and SSS to Table 2. We have also added references to the RMSE to Section 3.1.1 as follows:

Both CM2 configurations have a warm global mean SST bias (Table 2, 0.7° C for CM2-025 and 0.21° C for CM2-1) with a similar spatial pattern (Fig. 2a and c, also reflected in the similar root mean squared error (RMSE) between the models and observations, see Table 2).

The biases of the forced simulations have a similar pattern but are noticeably muted (Table 2, -0.09° C for OM2-025 and -0.06° C for OM2-1, with an almost halved RMSE) as the ocean surface is directly exposed to the fluxes based on the reanalysis product and is therefore close to observations.

The simulated global mean sea surface salinity (SSS) in CM2-1 matches well the observed estimates (0.02, Table 2), while CM2-025 is biased fresh (-0.1). Similar to SST, however, the global mean hides large spatial patterns of sizeable magnitude revealing problems of both models to adequately simulate various dynamical features (Fig. 3, RMSE in Table 2).

While they have a larger global mean SSS bias compared to the fully coupled models, this number is dominated by the Arctic Ocean and the RMSE is therefore much reduced (Table 2, Fig. 3b).

**Reviewer 2:** L206-208: I'm confused by this argument. The Southern Ocean SST bias is positive, which should reduce the equator-to-pole temperature gradient, reduce the meridional density gradient, and weaken the thermal wind contribution. Please clarify this line of argumentation.

**Reply**: The larger SST bias is indicative of stronger vertical stratification which in turn results in a faster flow. (Alternatively, one can think about this as the gradient at the southern ACC flank (i.e., ACC to pole gradient) increases.) We have added this information to the sentence as follows:

The larger transport of 211 Sv in CM2-025, along with a larger SST bias in the Southern Ocean which increases the vertical stratification, is consistent with a stronger thermal wind contribution to zonal velocity (Sect. 3.1.1, Fig. 2).

Reviewer 2: L210: "this type of model"

Reply: Fixed.

**Reviewer 2:** L232: What temporal resolution data is used to compute this standard deviation, for both the models and observations?

**Reply**: We use monthly data for both the model and observations. We have added the information on the temporal resolution to the text:

The standard deviation of monthly sea level anomaly serves as a good proxy for eddy activity and provides the possibility for comparison with an observational product.

**Reviewer 2:** Figure 6: Suggest adding an annotation to the figure (maybe a vertical line at 30S) to make it clear that two different zonal regions are being plotted on the same figure.

**Reply**: We have added a thick vertical line as well as text annotations to each panel to highlight the two different regions.

**Reviewer 2:** L330: What is the observational target for AABW density? Or it this statement derived from the T/S plots in Fig. 7?

**Reply**: The statement is derived from the temperature and salinity shown in Figure 8 (previously Figure 7). We have added a reference to the figure in the text:

CM2-025 forms noticeably denser AABW than observed with colder and saltier waters throughout the water column south of 55° S (Fig. 8a and b).

**Reviewer 2:** Figure 7: Suggest adding text annotations to panels 7a-h indicating the RMSE of these plots. This would provide a useful summary statistic for quickly comparing the models.

**Reply**: We have calculated and added the RMSE to Figure 8 (previously Figure 7).

**Reviewer 2:** Figure 8: Suggest reformatting this figure to be one row and two columns to improve the aspect ratio of the panels.

**Reply**: We have changed the aspect ratio to improve readability of the figure.

**Reviewer 2:** Lines 376-384: Does the sea ice concentration field in the Ross Sea show an imprint of the deep oceanic convection that occurs in this region? It is not visible in the sea ice extent, but I wonder if a patch of reduced sea ice concentration is visible. If so, it would be worth adding a comment on this.

Reply: While sea ice concentration is reduced in some parts of the deep oceanic convection area (Figure R5), there is no clear imprint of the deep convection (shown by the black mixed layer depth contour) onto the sea ice. We have added a sentence to the paragraph highlighting that the ocean surface in the deep convection areas is still covered by sea ice (though the sea ice concentration is lower than in observations, compare Figure R5 and Figure 10), as this might not be expected:

Note, that the ocean surface in areas of deep convection (Fig. 5) is covered by winter sea ice in all simulations, even if the concentration is lower than observed (not shown).

**Reviewer 2:** Lines 397-398: Is this difference really being driven by ice export rather than differences in thermodynamic growth? This statement needs to be backed up by some analysis of the sea ice mass budget terms or the velocity fields. Or the term "export" could simply be removed from this sentence.

**Reply**: We have rephrased the sentence to:

The ACCESS-CM2 models also have too much sea ice in the North Pacific (Bering Sea and Sea of Okhotsk) while the sea ice edge is well captured by the forced ACCESS-OM2 models.

Reviewer 2: Line 412: "Sea of Okhotsk"

Reply: Fixed.

Reviewer 2: Figure 10: The strip of data in the Arctic (north of roughly 80N) looks strange. Are these missing data values?

**Reply**: Thank you for pointing this out. Yes, these are missing values. Figure 11 (previously Figure 10) has been updated with the shading removed.

**Reviewer 2:** Figure 11: It would be helpful to also see the ENSO signal along the west coast of South America. Suggest including more data, such that the white ocean regions in Figs. 11a-c also show regression values.

**Reply**: We decided to keep the current extent as it reflects the area relevant for the SST regression. The map boundaries are extended to include continents which provides spatial context for easier orientation.

Reviewer 2: L448: "...too far East..."

Reply: Fixed.

Reviewer 2: L449: "... extend too far..."

Reply: Fixed.

Reviewer 2: L458: The text annotated in the plot say 0.09 rather than 0.11. Which one is correct?

**Reply**: The RMSE value of 0.09 as given on the figures is correct. We have updated the text, thanks for spotting the inconsistency.

**Reviewer 2:** L460-465: It would also be useful to see the power spectrum of the Nino 3.4 SSTA. Is this also markedly improved in the CM2-025 model? I leave it up to the authors whether they want to add this to Figure 11.

**Reply**: We have attached a power spectrum calculated on these simulation as requested, which supports the message delivered in the manuscript (Figure R6). We have added this information to the text as follows:

This result suggests that CM2-025 produces ENSO variability with a life cycle much closer to the observations (supported by a Niño3.4 SST power spectrum, not shown), which also suggests a significantly reduced ENSO bienniality.

However, we choose not to include a power spectrum in the manuscript as sensitivity testing undertaken while creating the CLIVAR ENSO metrics package (Planton et al. 2021) show that the results were sensitive to small methodological choices. Additionally, it was very

difficult to determine a single, robust metric from these calculations to gauge performance. We utilise the CLIVAR ENSO metric devised to assess the ENSO lifecycle (https://github.com/CLIVAR-PRP/ENSO\_metrics/wiki/EnsoSstTsRmse).

**Reviewer 2:** L503-507: Please provide more details on this. What aspects of the CM2-1 climate mean state would make it less prone to multidecadal variability?

**Reply**: We have extended this paragraph and added a discussion on sea ice in the Labrador Sea being an indicator for the possibility that a model exhibits such multidecadal variability: Mehling et al. (2024) identify a bias in the winter sea ice mean state in the Labrador Sea as a necessary condition to enable strong AMOC variability. The positive sea ice bias in the Labrador Sea in CM2-1 is indicative that AMOC variability is unlikely in this configuration.

Reviewer 2: Section 4.2: Does this model exhibit any multidecadal variability in the Southern Ocean associated with deep convection in the Ross and/or Weddell Seas? Given the mixed layer depth biases in these regions, they could be a candidate for multidecadal polynya variability arising from a charge/discharge mechanism of subsurface temperatures. This arises in many models, for example the GFDL CM4.0 model (see Fig. 3 of Held et al. (2019), 10.1029/2019MS001829). I am curious whether you see this in any of the ACCESS configurations.

Reply: The evolution of a multidecadal variability related to deep convection is limited to the North Atlantic (Labrador Sea) in CM2-025. Deep convection in the Ross and Weddell Seas occurs regularly in the ACCESS models, even at 0.25°. Figure R2 shows the annual maximum MLD for CM2-1 and CM2-025 for both regions. A charge/discharge mechanism as described in Held et al. (2019) for the GFDL-CM4 model is not active in ACCESS-CM2. We have added a note on this to the end of section 4.2:

Lastly, the evolution of a multidecadal variability associated with deep convection in CM2-025 is limited to the North Atlantic (Labrador Sea). CM2-025 does exhibit regular deep convection in the Ross and Weddell Seas (Fig. 5, 6) but it does not result in multidecadal variability in the Southern Hemisphere as reported for other models (Held et al., 2019).

**Reviewer 2:** L 525: Change to: "For this metric, CM2-025 compares best of all four models..." to make it clear that this statement is referring to Figure 4.

Reply: Changed.

Reviewer 2: L 547: Suggest changing to: "affects the AMOC and global mean surface temperatures."

**Reply**: Changed.

Reviewer 2: L551-554: Are there any planned future uses of CM2-025 (e.g. CMIP simulations, initialized predictions, etc.)? If so, it would be interesting to mention those in this final paragraph.

**Reply**: We added a sentence:

Planned uses of CM2-025 include a pre-industrial simulation that can serve as a spin-up for historical simulations, future scenarios or sensitivity experiments.

**Reviewer 3**

Reviewer 3: This article summarizes a higher-resolution ocean counterpart to the ACCESS CM2 climate model. The original model simulation utilizes a 1 degree ocean grid spacing. The new simulation uses a refined grid with 0.25 degree grid spacing. This is reported as the only difference between the two models, such that the comparison demonstrates the impact of resolving part of the mesoscale eddy field with the finer grid-spacing of the 0.25 degree model. The results are presented for the coupled "CM" version alongside a forced ocean and sea-ice experiment "OM" for context of the role of coupling. Overall the impact of refined resolution is mixed, with several aspects of the simulation being improved as expected, but other important model biases either being unaffected (or even emerging) in the 0.25 degree grid. The results are valuable and yield new insight into the role of ocean resolution in coupled climate models, so I support publishing this manuscript.

The study is well written and the analysis highlights many relevant features of a finer resolution ocean and is clearly presented. While there are previous studies to compare eddy-permitting and eddy-parameterizing models in the literature, I agree with the authors' statement that it is still useful to document this model and report the impact of high-resolution in different models since it can be very model specific. The approach to present results from the CM experiment alongside the OM experiment also helps clarify the role of horizontal resolution in coupled vs forced ocean models. I have several comments that could be taken by the authors' to clarify and improve certain aspects of the analysis and discussion in a revision.

**Reply**: We thank the reviewer for the supporting words on our manuscript and appreciate the suggestions to further improve the manuscript.

**General comments**

Reviewer 3: I appreciate the brevity of this article to keep the message succinct, since the CM2-1 and OM2 series models are already published resources. However, I do think some additional explanation of the eddy closure framework adopted in this work would be incredibly useful for a reader and the messages here. Perhaps it can be briefly summarized in the text, but more detail in an appendix could be warranted? It would be most useful to summarize the scale-aware eddy parameterization, especially since it is not retuned in the CM2-025 experiments. This could aid some potential reflection on scientific benefits vs computational cost for resolving vs parameterizing eddies that would bolster the discussion section. It would also benefit the paper to mention all employed parameterizations that may be relevant for the results and biases highlighted in this work, including for surface ocean vertical mixing (KPP?), any submesoscale parameterization, and any other relevant included parameterizations that can affect the mesoscale eddy field (lateral viscosity, diffusivity, etc).

**Reply**: This information was also requested by other reviewers. In brief, we use a weak GM/Redi parameterisation at these eddy-permitting resolutions. We have added a new paragraph to the end of section 2.1 to include these details:

The primary sub-grid-scale parameterisations in ACCESS-CM2 and ACCESS-OM2 are similar at each resolution, including the submesoscale parameterisation (Fox-Kemper et al.,

2008), biharmonic isotropic Smagorinsky horizontal friction (Griffies and Hallberg, 2000) and KPP vertical diffusivity (Large at al., 1994). The background diffusivity is set to zero at 0.25°, due to excessive numerical mixing, while the Jochum (2009) latitudinally-dependent, depth-independent background vertical tracer diffusivity scheme is used at 1°. (Details of the implementation of these parameterisations are provided by Kiss et al. 2020). The parameterisation of mesoscale eddies is applied more judiciously, to account for the partial resolution of mesoscale features at 0.25°. The Redi diffusivity for lateral isopycnal mixing is set to a constant 600 m2s-1 at 1° resolution, but is scaled by the grid resolution relative to either the first baroclinic Rossby radius, or the equatorial Rossby radius for latitudes between ±5°N, with a maximum diffusivity of 200 m2s1. The GM diffusivity is depth-independent but varies laterally, scaled by a combination of an inverse Eady growth rate timescale and a length scale squared (50 km at 1° and 20 km at 0.25°), as well as the same grid scaling used in the calculation of Redi diffusivity. There is a maximum diffusivity of 600 m2s-1 at 1° resolution and 200 m2s-1 at 0.25° resolution. The reduced eddy parameterisation at 0.25° has been tuned in ocean-sea ice simulations with a biogeochemical model to optimise transport of tracers by the ocean's overturning circulation.

Reviewer 3: Somewhat related to Point 1, a comment on the lack of "retuning" of CM2-025 compared to CM2-1 may aid some of the discussion. E.g., CM2-1 probably included some parameter optimization at some point in its development that targeted specific aspects of the simulation with a 1 degree ocean (Bi et al. 2020 documented albedo settings that were considered for improving ice and SST climatologies). If I understood it that the mesoscale eddy closure employed here utilizes a resolution function, then perhaps the hypothesis is that the model should not need any retuning. But in principle resolving some eddies is likely superior to parameterizing all eddies, so that this isn't so straightforward to me. The clear advantage for not retuning the model here is that it allows for directly comparing the effect of only changing ocean resolution. But likely the mean bias comparisons (e.g., SST, etc.) could be quite sensitive to the lack of retuning, since CM2-025 is not likely optimized in this configuration. To clarify this comment, I don't mean to imply that it is necessary to retune the model for this work, it just may warrant some discussion.

Reply: We agree that a comment on retuning is warranted. The tuning of the ocean part of ACCESS-CM2-025 has been largely achieved by previous tuning of ACCESS-OM2-025. Additional tuning of topography in key straits (which emerges in coupled modes but is controlled by salinity restoring in ocean-sea ice configurations) was done, but these minor modifications tend not to alter the large-scale circulation. We have added the following statement to the first paragraph of the conclusion to highlight the changes in eddy parameterisation and to clarify the potential for further retuning:

Third, the mesoscale eddy parameterisation scheme is weakened to account for partial resolution of eddies. The lack of any other retuning suggests that, with further efforts, the fidelity of CM2-025 could be further improved.

**Specific comments**

**Reviewer 3:** L26: "with only few assumptions on the climate forcing".

There are many other assumptions in these models, so I'm not quite sure I understood what was intended by the statement. Perhaps clarify.

**Reply**: This sentence aims to point out that climate models rely much less on prescribed external forcing, e.g., compared with forced ocean models which rely on an atmospheric product as a boundary condition. We have reworded the sentence to explain the reduced need for boundary conditions:

This interaction between the different model components allows the simulated climate system to freely evolve in time with only few assumptions on prescribed boundary conditions.

Reviewer 3: L30: "Increasing the lateral grid resolution of the ocean model has a high priority as the ocean is a key component that links different parts of the climate system." This statement doesn't obviously justify increasing the ocean resolution instead of the atmosphere or adding complexity elsewhere/ensembles/etc. This could be worded differently to better emphasize why this work focused on refining the ocean resolution. Was there a specific application or result in mind for a 0.25 degree model?

**Reply**: We have added a sentence at the end of this paragraph but since the remainder of the introduction provides a detailed motivation for increasing the ocean grid resolution, we prefer to keep the motivation brief:

A higher ocean grid resolution is not only expected to allow for a more realistic representation of the ocean (mean state and variability) but also impact how well sea ice and the lower atmosphere are simulated.

Reviewer 3: L32: "coastline, passages, and narrow straits" Also representing narrow coastal shelves.

**Reply**: We added the mentioning of narrow coastal shelves to this sentence:

The ocean grid resolution affects not only geometric constraints on the representation of the coastline, passages, narrow straits, and narrow coastal shelves, but also impacts the ability of a model to simulate mesoscale dynamics.

Reviewer 3: L35: "as detailed further below"

Suggest to be more specific. In the intro? In this study?

**Reply**: We have added the information that the following paragraphs of the introduction include such information:

The explicit representation of the ocean mesoscale, such as eddies but also boundary currents and fronts, improves both the ocean mean state and climate variability as detailed in the following paragraphs.

**Reviewer 3:** Section 2: Please confirm that the ocean time steps are also independent of resolution, or give their values (or state if dynamic).

**Reply**: The ocean time steps is increased from 30 min in CM2-1 to 20 min in CM2-025. We have added this information at the beginning of the third paragraph in Section 2.1:

The model components are coupled three hourly using the OASIS-MCT coupler (Valcke et

al., 2015; Craig et al., 2017 with an ice-ocean time step of 30 min in CM2-1 and 20 min in CM2-025.

Reviewer 3: L120: "deepening of individual grid cells on the Antarctic continental shelf to avoid model crashes"

Do you understand why this is only an issue in the coupled model? The atmospheric coupled model resolution is much coarser than JRA, so I would have expected the OM model to experience stronger forcing. Is there some coupled feedback mechanism?

**Reply**: These model crashes only occur during isolated storm events. We believe the reason that this issue only emerges in the coupled model is because such storm events are not as strong in the JRA55-do product used to force the OM2 models. We have added this information to the text as follows:

Additional key locations also required revision including a change of the land mask around the Antarctic continental shelf to avoid model crashes during local storm events. Storms can occasionally grow stronger in the coupled system compared with the prescribed atmosphere in the OM2 models (see Section 2.2), hence it is not an issue in OM2-025.

Reviewer 3: L123: "as deepening of the gateways of the Mediterranean and Baltic Seas which otherwise salinify or freshen in the coupled model due to limited exchange between the respective marginal sea and the open ocean."

Please clarify if this makes the gateway depths or cross-sectional areas closer to the measured values?

**Reply**: The goal of deepening the gateways was to improve exchanges between the marginal seas and the open ocean and not necessarily to match observations. We have added a sentence to the text:

The grid cell depth was chosen primarily to improve representation of water mass exchange, rather than to enhance overall realism.

**Reviewer 3:** L130: "Using present-day forcing allows for comparison of the model output to observations which are lacking the required spatial coverage for the pre-industrial climate which is often applied for climate model control simulations."

An alternative approach would have been to include a PI control spin-up for each model, followed by historical simulations with observed 1850-present forcing. Any comments on the reason for using a present condition to fully spin-up the model instead? Is there any consequence for this choice to the comparison to obs (or do you expect considering the transience of the present day forcing to matter?).

**Reply**: We appreciate that multiple approaches exist when developing and evaluating a new model configuration. The present-day configuration allows for a more accurate evaluation of the simulated ocean against observations, and with the ocean-sea ice model, which are the foci of this manuscript. Oceanic observations are more robust for the present-day period compared with the full historical period.

Reviewer 3: L135: "that is representative of the mean state"

Any comments on what is missed by excluding transience in these runs? Such as the various

interannual atmospheric variabilities that may rectify on the mean conditions? Perhaps it makes it somewhat more consistent with the coupled runs in the sense of the fixed climate forcings, but probably this is complicated since the coupled runs can simulate interannual modes of variability.

**Reply**: The text aimed to describe the climate conditions of the repeat year forcing of the OM2 atmospheric forcing. We aimed to say that the chosen 12-month period is representative of the mean state, i.e. most neutral in terms of major climate modes of variability. We have rephrased the sentence for clarity:

OM2 is forced three hourly with a repeat year forcing (May 1990 to April 1991) representative for a time period when the major climate modes of variability are neutral (Stewart et al., 2020) and that is based on the JRA55-do v1.3 atmospheric reanalysis product (Tsujino et al., 2018).

We further acknowledge that the OM2 modes do not exhibit any interannual variability. We assume, however, that by using a 100-yr time period for the analysis, most signals from any interannual variability present in CM2 but not in OM2 will average out. We have added this information to the manuscript:

The analysis focuses on the last 100 years (model years 400-499, blue shading in Fig. 1), a period long enough to largely average out imprints of climate modes of variability present in the CM2 models but not in OM2.

Reviewer 3: L151: "The analysis is guided by known CM2-1 biases"

This "known CM2-1 biases" qualifier is used a few times to this point. It could help to include a paragraph earlier discussing these known biases that will be explored (and maybe also why this resolution is expected to improve the specific bias).

**Reply**: We now briefly list some of the key diagnostics at the end of the introduction: The analysis focuses on known biases in ACCESS-CM2 (e.g., surface temperature and salinity, sea ice) and aspects of the climate system that we expect to respond to enhanced ocean grid resolution (e.g., ocean horizontal and overturning circulation, ocean-atmosphere interaction).

**Reviewer 3:** Table 2: The mixed layer depth box regions could be shown in Figure 5 if it doesn't make it too busy, e.g., to help a reader see what region encompasses the ACC.

**Reply**: We have added black boxes to Figure 5 to indicate the regions for which maximum mixed layer depths are reported in Table 2. We have slightly changed the eastern boundary for the Labrador Sea region, the values reported in Table 2 for CM2-1 and OM2-025 have been updated accordingly.

**Reviewer 3:** Figure 2: Some integrated metrics should be included here in the figure or caption to aid the comparisons, e.g., mean and RMS bias, spatial correlations.

**Reply**: We have added the RMSE for both SST and SSS to Table 2. We have also added references to the RMSE in the text as follows:

Both CM2 configurations have a warm global mean SST bias (Table 2, 0.7° C for CM2-025 and 0.21° C for CM2-1) with a similar spatial pattern (Fig. 2a and c, also reflected in the

similar root mean squared error (RMSE) between the models and observations, see Table 2).

The biases of the forced simulations have a similar pattern but are noticeably muted (Table 2, -0.09° C for OM2-025 and -0.06° C for OM2-1, with an almost halved RMSE) as the ocean surface is directly exposed to the fluxes based on the reanalysis product and is therefore close to observations.

The simulated global mean sea surface salinity (SSS) in CM2-1 matches well the observed estimates (0.02, Table 2), while CM2-025 is biased fresh (-0.1). Similar to SST, however, the global mean hides large spatial patterns of sizeable magnitude revealing problems of both models to adequately simulate various dynamical features (Fig. 3, RMSE in Table 2).

While they have a larger global mean SSS bias compared to the fully coupled models, this

While they have a larger global mean SSS bias compared to the fully coupled models, this number is dominated by the Arctic Ocean and the RMSE is therefore much reduced (Table 2, Fig. 3b).

Reviewer 3: L180: "The simulated global mean sea surface salinity (SSS) in CM2-1 matches well the observed estimates (0.02, Table 2), while CM2-025 is biased fresh (-0.1). Similar to SST, however, the global mean hides large spatial patterns of sizable magnitude revealing problems of both models to adequately simulate various dynamical features (Fig. 3)." Would it be more appropriate to discuss the RMS biases rather than absolute given these caveats?

**Reply**: As mentioned for above comment, we now reference to the RMSE in the text, including in the sentence this comment is referring to.

Reviewer 3: Figures 2&3: The Caspian Sea is presumably masked out of the models? I suggest to mask them out in the bias map rather than showing it as zero bias. Similarly, I wonder if CM2-1 Black Sea is also masked out? It seems in general the land mask in the figures is probably not consistent with the land mask from the models, which results in several "white" missing values regions in all figures which may be confusing due to many colormaps also containing white. It may be cleaner to use the model's land mask for all figures and eliminate any correspondence between the colormap and the color of any remaining missing values.

**Reply**: We have masked the Caspian Sea; the Black Seas is masked in CM2-1 but not in the 0.25° grids. We now show the missing values in Figures 2 and 3 in dark grey to distinguish from zero values (shown in white) and land (lighter grey). We decided to keep the current land mask to help the comparison between the models/panels.

Reviewer 3: L192: "in CM2-025 is improved in the Baltic and Mediterranean Seas" It has improved compared to what? The Baltic appears to have significant fresh bias in Figure 3a compared to 3b. I presume that is due to the salt restoring in 3b, but it seems awkward to say it is improved without some context. I wonder if there are similar strait transport issues in other seas with narrow exchanges that were not adjusted in this new grid, like the Black Sea and Persian Gulf?

**Reply**: We agree the text was not precise, we aimed to say that the new bathymetry product improved the salinity in the Baltic and Mediterranean Seas in CM2-025 (compared with

an older version of CM2-025). While the changes improved the salinity in these areas in CM2-025, the SSS biases remain large (CM2-025: 2.6 for Baltic Sea, 1.1 for Mediterranean). We have not adapted the bathymetry product in other marginal seas but have amended the text to mention potential benefits:

Other challenging regions to simulate are the marginal seas where there is limited exchange with the open ocean resulting in a delicate balance between exchange, precipitation, evaporation and river runoff. In particular, the salinity in CM2-025 in the Baltic and Mediterranean Seas did not stabilise when developing CM2-025. The CM2-025 topography has therefore been deepened to ensure sufficient exchange between the open ocean and the marginal sea. While this approach avoids a drift in salinity, the salinity biases remain large (Figure 2a). Other regions such as the Black Sea or the Persian Gulf could potentially benefit from similar changes to the bathymetry product but these were not considered necessary. The issue does not emerge in the forced models due to large surface salinity restoring in these regions (not shown).

Reviewer 3: L194: "modest sea surface salinity restoring" You could give the restoring piston velocity.

**Reply**: We have added more details on the surface salinity restoring, including the piston velocity, to Section 2.1:

The sea surface salinity in ACCESS-OM2 is globally relaxed toward the World Ocean Atlas 2013 v2 (WOA13, Zweng et al. 2013, same as used for initial conditions, see Section 2.2) through a salt flux, with the strength determined by a piston velocity of 33 m per 300 d (Kiss et al., 2020). The restoring salt flux is derived from the difference between surface salinity in the model and in WOA13 and kept below  $\pm 0.5$  psu to prevent excessively large fluxes.

**Reviewer 3: L261: "nearly identical"**

Please clarify the differences. Treguier et al. (2023, https://doi.org/10.5194/gmd-16-3849-2023) noted many important differences that can arise from inconsistencies, so that it can be difficult to compare MLD metrics if not 100% like-to-like model:obs definitions.

**Reply**: Assuming buoyancy is given as  $\Delta b = \frac{g\Delta\rho}{\rho_0}$  with  $g = 9.8 \text{ m s}^{-2}$ ,  $\rho_0 = 1035 \text{ kg m}^{-3}$  and  $\Delta b = 0.0003 \text{ m s}^{-2}$ , the buoyancy-based definition gives  $\Delta \rho = 0.0317 \text{ kg m}^{-3}$ . This is within 5% of the density definition used in the observational product. We have updated the text to report this number:

The buoyancy-based definition used in the models approximates the density-based definition used in the observational product to within 5%. For the purpose of this model validation, where we focus on the bias of the maximum MLD, the errors introduced due to the different choice of MLD definition is secondary as the models present large deep biases and observations tend to overestimate rather than underestimate MLD (Treguier et al., 2004).

Reviewer 3: L291: "All configurations overestimate the maximum MLD which excludes the atmospheric forcing to be primary source for the biases."

It isn't obvious to me that you can rule out the possibility that both the JRA repeat year forcing and coupled model are not good representations of the important features of the atmospheric

state. I tend to agree that a significant fraction of the bias is likely originating in some part of the ocean component, but it may not be the whole story and JRA can certainly have its own issues.

Reply: We agree that the JRA forcing might introduce some of the biases, which are possibly exaggerated by the infinite heat capacity of the forced OM2 models. However, we do not believe that the JRA forcing is the main source of error for the deep MLD in the OM2 models. As mentioned in the text, a version of OM2 at 0.1° resolution (forced by JRA) is able to simulate much more realistic max MLD. We have softened the wording to:

All configurations overestimate the maximum MLD which suggests the atmospheric forcing is less likely to be primary source for the biases.

Reviewer 3: L296: The deep MLD bias source could easily be caused by much more than just the vertical mixing scheme, it is likely to also be impacted by other processes and parameterizations that set the interior stratification. Notably, the chosen eddy parameterizations for meso and submesoscale eddies strongly impacts the wintertime stratification/restratification. Perhaps mention this here as well.

**Reply**: We have added the information to the text:

Improving the representation of MLD might therefore require a much higher grid resolution in both the horizontal and vertical (Kiss et a., 2020) as well as improvements to the vertical mixing schemes and mesoscale and submesoscale parametrisations which help to set the interior stratification by re-stratifying the water column in winter.

Reviewer 3: L296: Any comments on the summertime mixed layer depths? Are they not sensitive to the resolution?

**Reply**: We added a new Figure 6 (Figure R1 in response document) showing seasonal evolution of the max MLD at key locations (from Table 2). We have added a discussion of the MLD seasonality to the text throughout Section 3.1.3 as follows:

Figure 6 shows the seasonal evolution of the maximum monthly climatological MLD for selected regions.

Both the coupled and uncoupled models overestimate the maximum MLD in regions of deep MLD while summertime values are comparable (Fig. 5a-c, Fig. 6a-c,g-h).

Both OM2 models exhibit a deeper maximum wintertime MLD between June and November. The deepening occurs within about a month and then remains relatively constant through winter until November. In contrast, the CM2 models display a more gradual (and realistic) deepening, reaching their maximum in October. The faster MLD deepening in the OM2 models, which is not limited to the MLD evolution in the Weddell and Ross Seas (Fig. 6), likely results from the infinite heat capacity of the prescribed atmosphere, which enables rapid ocean heat loss. The MLD evolution is slower in the CM2 models as the atmosphere must adjust to the heat flux from the ocean.

The CM2 models exhibit slight improvements both for the peak wintertime MLD value as well as for the seasonal evolution.

**Reviewer 3:** L314: Some more insight for these AMOC calculations would help here. E.g., do these overturning strengths include any parameterized transports from the eddy closures?

Is the model the maximum overturning in density space (as per the figures) and can you please clarify that it is like-to-like with this observational estimate (it wasn't obvious to me from the reference that the quoted obs overturning value was computed in density space). Do the obs provide an error estimate? Are you also able to provide the density and/or depth of the peak overturning to compare with the obs?

You might include in this discussion - there is a potential role for specifics of the vertical coordinate along with the horizontal resolution for improving deep convection and overturning watermass pathways (e.g., as discussed by Wang et al., 2015 in the context of a z\* vs density based ocean component in an otherwise identical GFDL CM2 models, https://doi.org/10.1016/j.ocemod.2014.12.005; CM2M used MOM4, but I think shares several similarities with the MOM5 configuration here).

**Reply**: The overturning includes the GM component, we have added this information to the caption of Figure 7:

Zonally integrated meridional overturning circulation, including the transport from the eddy closure schemes, on potential density surfaces referenced to 2000 dbar for the Southern Ocean (SO) and Atlantic ocean basin, separated at 34° S as highlighted by the black vertical line, in (a) CM2-025, (b) OM2-025, (c) CM2-1, and (d) OM2-1.

We have added the error estimate of the observations (0.9 Sv) to Table 2.

All reported model values are the maximum overturning streamfunction at 26° N, calculated in density space. The peak transport occurs in the observations at 1030 m depth (RAPID, 2004-2012). We additionally converted the model overturning from density to depth space to extract the depth of the peak transport and compare with the available observations. The depth of the peak transport is biased shallow in the coupled models (782 m in CM2-025, 969 m in CM2-1), it matches the observed depth well in OM2-025 (1030 m) and is biased deep in OM2-1 (1173 m). We have added this information to the second paragraph of Section 3.1.4 as follows:

The strength of the upper cell at 26°N can be compared to available observations (McCarthy et al., 2015) by calculating the maximum transport in the vertical (where density is the vertical coordinate for the models and depth is the vertical coordinate for the observations). The observational estimate of 17.2 Sv is underestimated in all ACCESS configurations (Table 2).

Depth estimates of the simulated maximum transports are obtained by converting the model overturning from density to depth coordinates. The observed peak transport occurs at a depth of 1030 m and is biased shallow in the coupled models (782 m in CM2-025, 969 m in CM2-1), matches the observed depth well in OM2-025 (1030 m) and is biased deep in OM2-1 (1173 m).

We agree that there is scope to improve overturning pathways using alternative vertical coordinates, but that is not possible in MOM5 and is beyond the scope of this manuscript.

**Reviewer 3:** Figure 10: What is the red/orange stripe at the northern edge? Maybe a plotting artifact?

**Reply**: Thank you for pointing this out. The red/orange stripe at the northern edge are missing values. Figure 11 (previously Figure 10) has been updated with the shading removed.

Reviewer 3: L359: "The JRA55-do reanalysis product of the ACCESS-OM2 models" This wording confused me at first, it may be clearer to say "The JRA55-do reanalysis product used to force the ACCESS-OM2 models."

**Reply**: We changed the sentence to:

The JRA55-do reanalysis product used to force the ACCESS-OM2 models incorporates observed sea ice observations and has a strong imprint onto the simulated sea ice (Kiss et al. 2020).

**Reviewer 3:** S3.3: Can you also provide any comments on the impact on precipitation, if there is any impact?

**Reply**: We acknowledge it would be interesting to analyse the impact of the changed evaporation on the precipitation field but have decided to not include this additional evaluation in order to keep the manuscript concise.

Reviewer 3: L459: "This result suggests that CM2-025 produces ENSO variability with a life cycle much closer to the observations"

Figure 11 clearly demonstrates the reduced biennial variability. A power spectrum of SST variance, e.g. in the NINO3 region, might aid this discussion if there is more variability to show in the longer ENSO periods. (e.g., as in Figure 2 of https://doi.org/10.1029/2009GL038710)

**Reply**: We have attached below a power spectrum calculated on these simulation as requested (Figure R6), which supports the message delivered in the manuscript. We have added this information to the text as follows:

This result suggests that CM2-025 produces ENSO variability with a life cycle much closer to the observations (supported by a Niño3.4 SST power spectrum, not shown), which also suggests a significantly reduced ENSO bienniality.

However, we choose not to include a power spectrum in the manuscript as sensitivity testing undertaken while creating the CLIVAR ENSO metrics package (Planton et al. 2021) show that the results were sensitive to small methodological choices. Additionally, it was very difficult to determine a single, robust metric from these calculations to gauge performance. We utilise the CLIVAR ENSO metric devised to assess the ENSO lifecycle (https://github.com/CLIVAR-PRP/ENSO\_metrics/wiki/EnsoSstTsRmse).

Reviewer 3: Section 4.2: Is there any connection to the Southern Ocean polynas, besides simply being in regions of bottom water formation? I understand the mechanisms as hypothesized here are (mostly) local, but it is at least curious that these strong multi-decadal oscillations in deep convection regions occur in both high latitude regions in this model. You may at least elaborate on the Southern Ocean polynya timescales in this discussion and contrast the two high-latitude variabilities.

**Reply**: The evolution of a multidecadal variability related to deep convection is limited to the North Atlantic (Labrador Sea) in CM2-025. Deep convection in the Ross and Weddell Seas occurs regularly in the ACCESS models, even at 0.25°. Figure R2 shows the annual maximum MLD for CM2-1 and CM2-025 for both regions. At the end of Section 4.2, we now mention that deep convection in the Weddell and Ross Seas occurs regularly and is not

subject to multidecadal variability:

Lastly, the evolution of a multidecadal variability associated with deep convection in CM2-025 is limited to the North Atlantic (Labrador Sea). CM2-025 does exhibit regular deep convection in the Ross and Weddell Seas (Fig. 5, 6) but it does not result in multidecadal variability in the Southern Hemisphere as reported for other models (Held et al., 2019).

**Grammar**

Reviewer 3: L448: "displays the maximum ENSO SSTA to far east"

To -> too

 $\mathbf{Reply} \colon \mathrm{Fixed}.$

**Figures**

Figure R1: [New Figure 6 in main manuscript.] Seasonal evolution of maximum monthly climatological mixed layer depth in different regions for observations (De Boyer Montegut., 2023, black), CM2-025 (dark blue), CM2-1 (light blue), OM2-025 (dark red), and OM2-1 (light red). The different regions are shown as black boxes in Figure 5.

Figure R2: Maximum mixed layer depth in a year for the Weddell and Ross Seas in CM2-025 (dark blue) and CM2-1 (light blue).

Figure R3: Zonal annual average evaporation bias  $(mm \, day^{-1})$  of CM2-025 and CM2-1 relative to IFREMER v4.1 (1992-2018).

Figure R4: Sea surface and interior temperature drift in CM2-025 under present-day (blue) and pre-industrial (orange) conditions.

Figure R5: September sea ice concentration (shading) and 1000-m mixed layer depth (black contour) for the Southern Hemisphere in the CM2 and OM2 models.

Figure R6: Nino 3.4 sea surface temperature power spectrum for CM2-1, CM2-025, and the  ${\it HadISST}$  observational product.

---

## Author Response (AR2)

**Response to reviewer comments**

We sincerely thank the editor, Riccardo Farneti, and the anonymous reviewer for their efforts to thoroughly review our revisions to the original manuscript. We detail here responses to the final outstanding comments and indicate changes made to the manuscript. The reviewers' comments have been listed in order below in italic text, followed by our responses in normal text, and new manuscript text in blue.

**Reviewer 1**
* * *
**Reviewer 1:** *The authors have addressed all previously raised points, and the manuscript has improved considerably. I have one additional minor consideration that requires clarification before I can recommend acceptance.*

**Reply**: Thank you for these encouraging remarks in support of this manuscript.
* * *
**Reviewer 1:** *Mixed-layer depth definition and comparison with observations*
*The discrepancy between model and observational MLD arises not only from the different threshold criteria but also from differences in the reference depth used in each calculation. The new observational MLD product (de Boyer Montégut et al., 2023) defines MLD relative to 10 m, whereas the model computes MLD relative to the depth of the first model layer. This distinction can have substantial implications for the comparison. As noted by Treguier et al. (2023), "a difference of less than 10 m in the reference depth can lead to more than 40 m difference in the MLD climatology." I suggest that the authors explicitly discuss this additional source of discrepancy in the text around line 311.*

**Reply**: We have added this information to the manuscript as follows: Further, a discrepancy of more than 40 m (Treguier et al., 2023) in estimated MLD may exist due to the different reference depths used in the observational product (10 m) and in the model (surface grid cell).

**Typos**
* * *
**Reviewer 1:** *Line 256 – add a space before "along."*

**Reply**: Fixed (at line 245).

***Reviewer 1:*** *Figure 5 – add a period after "Figure 6."*

**Reply**: Fixed.
* * *
***Reviewer 1:*** *Lines 631–632 – remove the second occurrence of "can serve as," as it appears twice.*

**Reply**: Removed.